# Quantum Robust Inner Minimization for Reinforcement Learning with Quadratic Speed-Up in Query Complexity

**Hyun Kyu Lee** [1]  **Joongheon Kim** [2]  **Sung Whan Yoon** [1] [3]

## Abstract

Robust reinforcement learning (RRL) aims to tackle unexpected environmental changes by optimizing policies against the worst case. However, RRL remains impractical due to the cost of the Max-Min optimization, where it suffers from the exhaustive query complexity for finding the worst-case (dubbed 'Min') within the environmental uncertainty set $\mathcal{U}$, i.e., $\mathcal{O}(|\mathcal{U}|)$. By viewing this via a lens of quantum perspective, we raise a pivotal question: *If we can query from the environment with quantum superpositions, is it possible to accelerate the Max-Min optimization of RRL?* Our answer is 'Yes'. Our method, called quantum robust inner minimization (QRIM), encodes the uncertainty set with quantum superposition and amplifies low-return cases, thus enabling RL for solving the robust (i.e., worst-case) Bellman equation. Importantly, QRIM achieves a quadratic speed-up in query complexity without altering the outer RL pipeline, i.e., $\mathcal{O}(\sqrt{|\mathcal{U}|})$. Validated through classical simulations to real quantum hardware execution, QRIM learns more robust policies with quadratically reduced queries than classical RL.

## 1. Introduction

Reinforcement learning (RL) optimizes agents' actions in a given environment by maximizing the expected return, where the parameterized rule for taking an action is called *policy*. In practical scenarios, RL often suffers from a drastic performance degradation when deployed in a real-world environment, which shows a significant gap with simulations. Robust reinforcement learning (RRL) primarily aims

to address this mismatch by introducing an *uncertainty set*, which covers possible environment variations, and maximizing the policy's performance (dubbed 'Max') under the worst-case scenario (dubbed 'Min'), so-called 'Max-Min' optimization. By assuming the state-action rectangular uncertainty, the robust value function satisfies robust Bellman equations, and it permits agents to find a principled planning procedure in a finite space (Iyengar, 2005; Nilim & Ghaoui, 2005; Wiesemann et al., 2013).

The existing RRL algorithms approximate their robust policy solutions by applying the robust Bellman operator at each sampled state–action pair, iteratively. In each step, RRL requires solving the inner minimization via recognizing the worst-case transition within the uncertainty set, i.e., $\mathcal{U}$, and estimating the corresponding Bellman backup. The crucial problem is that when the uncertainty set expands, or a tighter confidence guarantee of the approximation is required, the inner minimization process, aimed at searching for the worst-case, rapidly inflates the computational and query (i.e., sample) complexities, whose complexity can be approximated as $\mathcal{O}(|\mathcal{U}|)$. This becomes the primary bottleneck of practical usage of RRL, regardless of whether the RL architecture follows value-based or policy gradient.

We aim to view this through a quantum perspective to anticipate a novel remedy for the bottleneck. Let us begin with a pivotal question: *If queried from the environment with quantum superpositions, can we accelerate the inner minimization process of RRL?* To represent a given environment with superposition, we borrow the concept of *quantum-accessible environment* (Wang et al., 2021), which is an interface that prepares a quantum superposition over possible next states with amplitudes equal to the square roots of their transition probabilities. Specifically, for a state–action pair $(s, a)$ and each uncertain case in $\mathcal{U}$, a quantum-accessible environment provides *unitaries* $U_P$, $U_R$, and $U_V$ with quantum superpositions over next states and loads the associated rewards and values (see Definition 3.3; subscripts $P$, $R$, $V$ respectively denote probability, reward, and value); thereby coherently encoding the one-step return on an ancilla for amplitude estimation. Built on it, our quantum robust inner minimization (QRIM) estimates the amplitude of the worst case, which has the minimal reward within the uncertainty set by employ-

[1]Graduate School of Artificial Intelligence, Ulsan National Institute of Science and Technology (UNIST), South Korea. [2]School of Electrical Engineering, Korea University, South Korea. [3]Department of Electrical Engineering, UNIST, South Korea. Correspondence to: Sung Whan Yoon <shyoon8@unist.ac.kr>.

*Proceedings of the 43$^{rd}$ International Conference on Machine Learning*, Seoul, South Korea. PMLR 306, 2026. Copyright 2026 by the author(s).

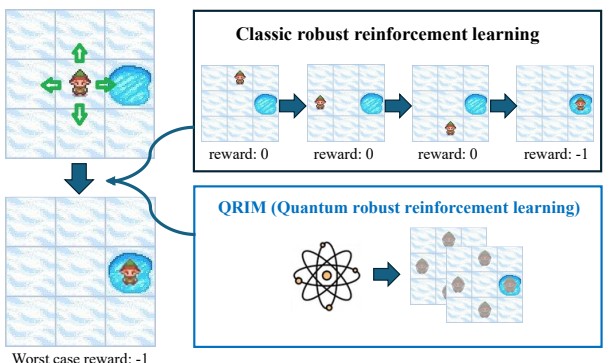

*Figure 1.* **Query complexity in a toy MDP** ($|\mathcal{U}|$=4). **(Top)** Classical RRL requires $\mathcal{O}(|\mathcal{U}|)$ sequential scans. **(Bottom)** QRIM uses superposition, searching for the worst-case in $\mathcal{O}(\sqrt{|\mathcal{U}|})$ steps.

ing a quantum minimum finding algorithm, whose query complexity is $\mathcal{O}(\sqrt{|\mathcal{U}|})$, which is significantly lower than the conventional worst-case exhaustive search (i.e., $\mathcal{O}(|\mathcal{U}|)$ of RRL). We emphasize that a quantum approach can alleviate the impracticality of sample complexity for classical robust policy training, shedding light on a novel benefit of rethinking RRL problems from a quantum perspective.

To intuitively illustrate the speed-up gains of our approach, we present a toy example inspired by a local snapshot of the FrozenLake environment in Figure 1, which is a navigation problem. Consider an agent facing an uncertainty set of size $|\mathcal{U}| = 4$ (e.g., four possible wind directions), where one direction leads to a hole representing the worst-case scenario with a negative reward ($-1$), while others are safe (0). In the classical robust RL framework (at the top box in Figure 1), the agent must sequentially simulate each of the four transition possibilities to identify the minimum reward. This exhaustive linear scan results in a query complexity of $\mathcal{O}(|\mathcal{U}|)$. In contrast, QRIM (at the bottom box in Figure 1) encodes these scenarios into a quantum superposition, visualized as transparent grids processed simultaneously. QRIM identifies the worst-case outcome with a quadratic speed-up, requiring only $\mathcal{O}(\sqrt{|\mathcal{U}|})$ queries, which is 2.

Our contributions are summarized as follows:
- We propose QRIM that accelerates inner minimization of RRL by encoding uncertainty sets in quantum superposition and amplifying low-return cases.
- We provide theoretical guarantees on QRIM's accuracy and query complexity, showing a quadratic speed-up over classical RRL methods.
- In a classical emulator, QRIM learns robust policies with significantly reduced query counts: $\times 2.17$ out-of-distribution reward over classical RL and $-79.5\%$ query reduction over classical RRL in the CartPole benchmark.
- We validate QRIM on real quantum hardware, i.e., IBM 127-qubit system, to exhibit consistent robustness and query efficiency even under quantum noise.

## 2. Related Work

### 2.1. Robust reinforcement learning

Robust Markov decision processes (MDPs) introduce rectangular uncertainty sets at the state–action level and show that the associated robust Bellman operator is a contraction with a unique fixed point in finite spaces (Iyengar, 2005; Nilim & Ghaoui, 2005); these results underpin robust planning by value or policy iteration (Wiesemann et al., 2013). There exist two dominant families of RRL methods: value-based learners that approximate targets induced by the robust Bellman operator (Iyengar, 2005; Nilim & Ghaoui, 2005) and policy-gradient learners that derive gradients for the Max-Min objective under rectangular uncertainty (Wang & Zou, 2022; Kumar et al., 2023). In both cases, each update at a sampled state–action pair requires the *inner minimization* (so-called worst-case Bellman backup), which becomes the dominant cost as the uncertainty resolution grows. To mitigate this cost, adversarial training learns a disturbed policy to intentionally expose agents to the failure cases (RARL) (Pinto et al., 2017), and ensemble methods optimize lower quantiles over environment samplings (EPOpt) (Rajeswaran et al., 2017); these approaches aim to mimic the undesired defectives in training yet avoid the explicit worst-case backup. A complementary direction replaces the uncertainty set with the set-valued kernels by distributional ambiguity balls bounded by total variation or Wasserstein distance, yielding geometry-specific convex inner problems and generalization bounds (Xu & Mannor, 2010; Smirnova et al., 2019; Goyal & Grand-Clement, 2023), particularly with accelerated solvers for the Wasserstein case (Clement & Kroer, 2021; Yu et al., 2023). Nonetheless, evaluation on fine-grained uncertainty grids still requires $O(|\mathcal{U}|)$ scans over candidate scenarios.

### 2.2. Quantum reinforcement learning

Quantum reinforcement learning (QRL) has developed along two main streams (Meyer et al., 2022). The first employs parameterized quantum circuits (PQCs) to parameterize policies or value functions, trained with classical optimizers. Jerbi et al. (2021) introduced PQC-based policies and demonstrated their trainability on RL environments with a few qubits, highlighting potential benefits. In parallel, Kim et al. (2025) explored PQC policies in applied control settings, emphasizing how circuit design choices and task structure influence learning stability and performance. The second direction shifts its focus to an environment perspective, whose main goal is to construct an oracle-like access to a quantum-accessible environment (or quantum generative model). The core benefits are the accelerated planning or evaluation by composing coherent state preparation with standard quantum primitives. As a key enabler for estimating value functions in a quantum-accessible environment,

Wang et al. (2021) introduced algorithms that leverage quantum amplitude estimation (QAE) to reduce the dependency on the target accuracy compared to classical Monte Carlo estimation. QAE estimates a bounded mean within additive error $\varepsilon$ using $\mathcal{O}(\varepsilon^{-1})$ queries compared to the $\mathcal{O}(\varepsilon^{-2})$ calls needed by classical Monte Carlo (Brassard et al., 2000).

Despite these advances, robustness has not been a primary focus in QRL: variational approaches study the expressivity and learnability of quantum policies, while oracle-based works accelerate classical planning or exploration (Xu & Aggarwal, 2025; Ganguly et al., 2025); neither directly targets the Max-Min structure of robust MDPs or accelerates repeated inner minimization that overloads training. In contrast, our method, i.e., QRIM, computes robust Bellman backups across a discretized uncertainty set and identifies the worst case, thereby achieving robustness via a quantum-accessible framework.

# 3. Preliminaries

As preliminaries, let us begin with providing standard notations for RL and proceed to formally describe robust MDPs with rectangular uncertainty sets. Afterward, for a quantum perspective, let us specify the formal definition of a quantum-accessible environment and its basic propositions.

## 3.1. RL background

Let $\mathcal{S}$ and $\mathcal{A}$ be the state and action spaces, and let $P$ denote the transition kernel. For each state–action pair $(s, a)$ with $s \in \mathcal{S}, a \in \mathcal{A}$, the state transition probability is $P(\cdot \mid s, a) \in \Delta_{\mathcal{S}}$, where $\Delta_{\mathcal{S}}$ is the set of probability measures over $\mathcal{S}$. An MDP $(\mathcal{S}, \mathcal{A}, P, r, \gamma)$ with discount factor $\gamma \in (0, 1)$ evolves with a state transition from step $t$ to $t + 1$, i.e., $s_{t+1} \sim P(\cdot | s_t, a_t)$, and returns a reward, i.e., $r_t = r(s_t, a_t, s_{t+1})$. A stationary policy $\pi(\cdot | s)$ induces value functions:

$$V^\pi(s) = \mathbb{E}_\pi\Big[\sum_{t \geq 0} \gamma^t r_t \,\Big|\, s_0 = s\Big],$$
$$Q^\pi(s, a) = \mathbb{E}_\pi\Big[\sum_{t \geq 0} \gamma^t r_t \,\Big|\, s_0 = s, a_0 = a\Big].$$

The Bellman operators are formulated as follows:

$$(T^\pi V)(s) = \mathbb{E}_{a \sim \pi}\mathbb{E}_{s' \sim P}\big[r(s, a, s') + \gamma V(s')\big],$$
$$(T_* V)(s) = \max_{a \in \mathcal{A}} \mathbb{E}_{s' \sim P}\big[r(s, a, s') + \gamma V(s')\big].$$

Both are $\gamma$-contractions in $\|\cdot\|_\infty$, yielding unique fixed points and supporting value/policy iteration in finite spaces. With a function approximation, *value-based* learners estimate sampled Bellman targets $r(s, a, s') + \gamma V(s')$, while *policy-gradient* learners optimize $J(\theta) = \mathbb{E}[V^{\pi_\theta}(s)]$ via computing gradients:

$$\nabla_\theta J(\theta) = \mathbb{E}_{(s,a) \sim d^{\pi_\theta}}\big[\nabla_\theta \log \pi_\theta(a \mid s)\, A^{\pi_\theta}(s, a)\big],$$
$$A^\pi(s, a) = Q^\pi(s, a) - V^\pi(s).$$

Especially to mitigate the high variance of policy gradients, actor–critic methods introduce a learned value function to stabilize updates. In value-based and actor–critic methods alike, learning relies on estimating accurate Bellman targets for critic updates at sampled states.

## 3.2. Robust MDPs under rectangular uncertainty

In standard MDPs, the transition kernel $P(\cdot \mid s, a)$ and reward $r(s, a, s')$ are assumed to be fixed. Robust reinforcement learning instead considers a set of possible transition–reward pairs, called an *uncertainty set*, which reflects possible variations in transition kernels and rewards. At each state–action pair $(s, a)$, the adversary may choose one candidate model $(P, r)$ from this set, resulting in different one-step transition outcomes. To make this precise and to fix notation, we formalize the uncertainty at $(s, a)$ by an uncertainty set of transition–reward pairs and specify the notion of *rectangularity* as follows.

**Definition 3.1** (Uncertainty set and rectangularity (Iyengar, 2005; Nilim & Ghaoui, 2005)). *Let* $\mathcal{U}_{sa} \subseteq \{(P, r) : P(\cdot \mid s, a) \in \Delta_{\mathcal{S}}, \ r : \mathcal{S} \to \mathbb{R}\}$ *be a nonempty set of one step laws, where* $(P, r)$ *specifies both the transition kernel* $P(\cdot \mid s, a)$, *the reward function* $r(s, a, \cdot)$ *and* $\Delta_{\mathcal{S}}$ *is the set of probability measures over* $\mathcal{S}$. *Rectangular uncertainty allows the adversary to choose an element of* $\mathcal{U}_{sa}$ *independently at each* $(s, a)$.

For bounded rewards, the robust optimality operator $T_{\mathrm{rob}}$, which is the robust Bellman operator, takes the form of a Max-Min problem:

$$(T_{\mathrm{rob}} V)(s) = \max_{a \in \mathcal{A}} \min_{(P,r) \in \mathcal{U}_{sa}} \mathbb{E}_{s' \sim P(\cdot | s, a)}\big[r(s, a, s') + \gamma V(s')\big]. \tag{1}$$

**Proposition 3.2** (Contraction and fixed point (Iyengar, 2005; Nilim & Ghaoui, 2005)). *$T_{\mathrm{rob}}$ is a $\gamma$-contraction in $\|\cdot\|_\infty$ with unique fixed point $V^*$. Any policy greedy with respect to the robust action-value function $Q^{\mathrm{rob}}(\cdot, \cdot; V^*)$ is robust-optimal, where*

$$Q^{\mathrm{rob}}(s, a; V) = \min_{(P,r) \in \mathcal{U}_{sa}} \mathbb{E}_{s' \sim P(\cdot | s, a)}\big[r(s, a, s') + \gamma V(s')\big]. \tag{2}$$

We treat $\mathcal{U}_{sa}$ as a finite candidate set, which is the standard formulation of RRL with discrete uncertainty (Nilim & Ghaoui, 2005; Rajeswaran et al., 2017). Specifically, we discretize each $\mathcal{U}_{sa}$ into $N$ candidates $\{(P_i, r_i)\}_{i=1}^N$ (i.e., $|\mathcal{U}_{sa}| = N$, the per-state–action uncertainty set size), such as grids over parameters, ensembles, or samples from ambiguity balls (Xu & Mannor, 2010). The inner minimization for each candidate is then written as

$$g_i(s, a; V) := \mathbb{E}_{s' \sim P_i(\cdot | s, a)}\big[r_i(s, a, s') + \gamma V(s')\big],$$
$$Q^{\mathrm{rob}}(s, a; V) = \min_{i \in [N]} g_i(s, a; V). \tag{3}$$

The practical bottlenecks are identifying $Q^{\text{rob}}$ and accurately evaluating $g_i$ from many $(s, a)$ queries. An extension to continuous uncertainty sets is provided in Appendix C.6.

### 3.3. Quantum-accessible environment and basic propositions

We build upon the quantum-accessible environment in RL planning (Wang et al., 2021) and the basic propositions for amplitude estimation and minimum searching in quantum settings. We use standard quantum computing Dirac notation: kets such as $|x\rangle$ denote computational-basis states (e.g., $|00\rangle, |01\rangle, |10\rangle, |11\rangle$), and tuples such as $|x, y, z\rangle$ abbreviate the tensor product $|x\rangle \otimes |y\rangle \otimes |z\rangle$. A general quantum state takes the form $|\psi\rangle = \sum_x \alpha_x |x\rangle$, a superposition over basis states $|x\rangle$ with complex *amplitudes* $\alpha_x$. Operations on $|\psi\rangle$ are *unitary* (reversible, norm-preserving) linear maps, i.e., $|\psi\rangle \mapsto U |\psi\rangle$, and a *measurement* returns basis state $|x\rangle$ with probability $|\alpha_x|^2$. An *oracle* is a unitary; a *coherent query* applies it to the entire superposition, processing all basis states at once. *Amplitude amplification* then iteratively increases the amplitude of target basis states, so that the desired outcome can be measured with high probability. Additional explanation of the notation is in Appendix A.

**Definition 3.3** (Quantum-accessible environment). For classical inputs $(s, a)$ and a basis index $i \in [N]$ of a discretized environment, a quantum-accessible environment provides unitaries as follows:

(i) Transition sampler $U_P$ :
$$|s, a, 0\rangle \mapsto \sum_{s'} \sqrt{P_i(s' \mid s, a)} |s, a, s'\rangle,$$

(ii) Reward oracle $U_R$ :
$$|s, a, s', 0\rangle \mapsto |s, a, s', r_i(s, a, s')\rangle,$$

(iii) Value oracle $U_V$ :
$$|s', 0\rangle \mapsto |s', V(s')\rangle,$$

together with standard reversible arithmetic on fixed-precision registers (add/multiply/compare).

**Proposition 3.4** (Quantum amplitude estimation (Brassard et al., 2000)). *Given a unitary $A$ that prepares $\sqrt{\mu} |1\rangle + \sqrt{1 - \mu} |0\rangle$ on an ancilla (possibly entangled), there exists an algorithm that, with $M$ coherent queries to $A$ and $A^\dagger$, outputs $\widehat{\mu}$ such that $|\widehat{\mu} - \mu| \le c/M$ with probability at least $2/3$ for a universal constant $c$. Median-of-means boosting yields, for any $\varepsilon, \delta \in (0, 1)$, an estimate with $|\widehat{\mu} - \mu| \le \varepsilon$ and failure probability $\le \delta$ using $\mathcal{O}(\varepsilon^{-1} \log(1/\delta))$ queries.*

QAE estimates an amplitude-encoded mean $\mu$ to additive accuracy $\varepsilon$ using $\mathcal{O}(1/\varepsilon)$ coherent queries to the state-preparation unitary $A$, in contrast to the $\mathcal{O}(1/\varepsilon^2)$ samples required by classical Monte Carlo. This yields a quadratic improvement in the precision dependence.

**Proposition 3.5** (Quantum minimum finding (Durr & Hoyer, 1996)). *Let $f : \{1, \ldots, N\} \to \mathbb{R}$ be accessible through an oracle that, given two indices, can mark (or compare) the smaller value with error probability at most $\eta < \frac{1}{2}$. Then there is a procedure that returns $\widehat{i} \in \arg\min_i f(i)$ with failure probability $\le \delta$ using $\tilde{\mathcal{O}}(\sqrt{N} \log(1/\delta))$ mark/value queries; the overhead from $\eta$ can be absorbed into the polylog factor by repetition.*

Quantum minimum finding (QMF) repeatedly applies amplitude amplification to mark indices whose value falls below a current threshold, achieving $\mathcal{O}(\sqrt{N})$ query complexity compared to the $\mathcal{O}(N)$ of classical exhaustive search.

In our application, we set $f(i) = g_i(s, a; V)$. Since each $g_i$ is itself an expected value over the next-state distribution rather than a directly readable scalar, the inner minimization is realized by *composing* the two primitives above: **Proposition** 3.4 provides each candidate value $g_i$ via amplitude estimation, and **Proposition** 3.5 performs the search over these values through a coherent comparison oracle. The full construction of this composition is detailed in Section 4.2. We provide explicit circuit constructions of the underlying oracles for the RL environments used in our experiments in Appendix B, illustrating how the quantum-accessible environment can be realized in principle.

## 4. Methodology

Before describing our QRIM, we bound the one-step return $r_i(s, a, s') + \gamma V(s')$ and encode it into an ancilla amplitude, so that each candidate value $g_i$ can be recovered via QAE.

**Assumption 4.1** (Bounds for normalization). Rewards satisfy $R_{\min} \le r_i(s, a, s') \le R_{\max}$, and $V$ is bounded. Set $V_{\max} = R_{\max}/(1 - \gamma)$ and $V_{\min} = R_{\min}/(1 - \gamma)$, hence

$$L := R_{\min} + \gamma V_{\min}, \qquad U := R_{\max} + \gamma V_{\max},$$

so $r_i(s, a, s') + \gamma V(s') \in [L, U]$ for all possible outcomes.

Under Assumption 4.1, we rescale the one-step return to $u_i(s') \in [0, 1]$. Since $g_i$ is not a directly readable scalar but an expectation over $s'$, Proposition 4.2 encodes it into a quantum state in which the ancilla's $|1\rangle$ measurement probability equals $\mu_i = \mathbb{E}_{s' \sim P_i}[u_i(s')]$, which QAE estimates to prepare $g_i$.

**Proposition 4.2** (Amplitude encoding of $g_i$). *Under Definition 3.3 and Assumption 4.1, one can implement a state-preparation unitary $\mathcal{G}_i(s, a)$ for fixed $(s, a)$, which prepares*

$$\mathcal{G}_i(s, a)|0\rangle = \sum_{s'} \sqrt{P_i(s' \mid s, a)} |s'\rangle$$
$$\cdot \left( \sqrt{u_i(s')} |1\rangle + \sqrt{1 - u_i(s')} |0\rangle \right),$$

*with $u_i(s') = (r_i(s, a, s') + \gamma V(s') - L)/(U - L)$ where $u_i(s') \in [0, 1]$. Let $\mu_i = \mathbb{E}_{s' \sim P_i}[u_i(s')]$, then $\mu_i = (g_i - L)/(U - L)$.*

**Algorithm 1** Quantum-Accelerated Robust Reinforcement Learning

1: Initialize policy/value parameters; expose the classical value oracle $U_V$.
2: **repeat**
3:     Collect data (on-policy for policy-gradient, or from replay for value-based).
4:     **for** each sampled $(s, a)$ **do**
5:         $\widehat{Q}^{\mathrm{rob}}(s, a; V) \leftarrow \mathrm{QRIM}(s, a; V)$ {Algorithm 2}
6:     **end for**
7:     Form targets/advantages using $\widehat{Q}^{\mathrm{rob}}$ and update with a standard optimizer.
8: **until** convergence or budget

**Algorithm 2** QRIM: Quantum Robust Inner Minimization

  **Require:** Oracles $\{\mathcal{G}_i(s, a)\}_{i=1}^N$; tolerances $(\varepsilon_{\mathrm{cmp}}, \varepsilon_{\mathrm{est}})$; failure probability $\delta$
1: **function** $U_{\mathrm{ESTIMATE}}(i; \varepsilon)$: QAE on $\mathcal{G}_i(s, a)$ at accuracy $\varepsilon$; coherently writes $|i\rangle |0\rangle \mapsto |i\rangle |\widehat{g}_i\rangle$
2: **function** $U_{\mathrm{LESS}}(i, j)$: coherent phase oracle $|i\rangle |j\rangle \mapsto (-1)^{[\widehat{g}_i \leq \widehat{g}_j]} |i\rangle |j\rangle$
3:     Apply $U_{\mathrm{ESTIMATE}}(i; \varepsilon_{\mathrm{cmp}})$ and $U_{\mathrm{ESTIMATE}}(j; \varepsilon_{\mathrm{cmp}})$
4:     Apply reversible comparator with phase flip $(-1)^{[\widehat{g}_i \leq \widehat{g}_j]}$
5:     Apply $U_{\mathrm{ESTIMATE}}^{\dagger}(j; \varepsilon_{\mathrm{cmp}})$ and $U_{\mathrm{ESTIMATE}}^{\dagger}(i; \varepsilon_{\mathrm{cmp}})$ to uncompute the auxiliaries
6:   $\widehat{i} \leftarrow$ QMF over $i \in [N]$ with phase oracle $U_{\mathrm{LESS}}$
7: Apply $U_{\mathrm{ESTIMATE}}(\widehat{i}; \varepsilon_{\mathrm{est}})$ and **measure** the value register $\rightarrow \widehat{Q}^{\mathrm{rob}}(s, a; V)$
8: **return** $\widehat{Q}^{\mathrm{rob}}(s, a; V)$

Equipped with the amplitude encoding of Proposition 4.2, we here describe the inner minimization routine called QRIM, whose interface is compatible with value-based and policy-gradient updates. The proof is in Appendix C.1.

### 4.1. Main algorithm

As described in Algorithm 1, our quantum routine repeatedly returns an estimator $\widehat{Q}^{\mathrm{rob}}(s, a; V)$ for state–action samples, thereby converging to the robust policy. The essential part is executed by QRIM, which computes $\widehat{Q}^{\mathrm{rob}}(s, a; V)$ for each state–action pair.

### 4.2. Inner minimization: QRIM

For each discretized candidate environment $(P_i, r_i)$ at a given $(s, a)$, define

$$g_i(s, a; V) = \mathbb{E}_{s' \sim P_i}\big[r_i(s, a, s') + \gamma V(s')\big],$$
$$Q^{\mathrm{rob}}(s, a; V) = \min_{i \in [N]} g_i(s, a; V).$$

Under the quantum-access interface (**Definition** 3.3) and normalization (**Assumption** 4.1), **Proposition** 4.2 allows QRIM to provide a unitary $\mathcal{G}_i(s, a)$ whose ancilla amplitude has mean $\mu_i = \frac{g_i - L}{U - L}$. We utilize amplitude estimation (**Proposition** 3.4) as a building block and compose it with QMF (**Proposition** 3.5).

QRIM exposes two tolerances of minimum finding and amplitude estimation:

- $\varepsilon_{\mathrm{cmp}} > 0$ (comparison tolerance): a tolerance used when comparing two candidates inside QMF. Each comparison invokes QAE on $\mathcal{G}_i$ and $\mathcal{G}_j$ with an additive error $\varepsilon_{\mathrm{cmp}}$, deciding whether $g_i \leq g_j$ with high confidence.

- $\varepsilon_{\mathrm{est}} > 0$ (estimation tolerance): a tolerance used when re-estimating the winner, i.e., the estimated worst-case index $\widehat{i} \in [N]$ returned by QMF, at the end, producing the robust value $\widehat{Q}^{\mathrm{rob}}(s, a; V)$.

More sampling of coherent queries is required for reducing the uncertainties (**Proposition** 3.4). Figure 2 provides a structural overview, and Algorithm 2 shows the algorithmic description of QRIM. Briefly, Algorithm 2 defines a coherent estimator $U_{\mathrm{ESTIMATE}}$ (line 1) and a phase oracle $U_{\mathrm{LESS}}$ (lines 2–5) that compares candidates at tolerance $\varepsilon_{\mathrm{cmp}}$. QMF then uses $U_{\mathrm{LESS}}$ to identify the worst-case index $\widehat{i}$ (line 6), which is re-estimated by $U_{\mathrm{ESTIMATE}}$ at accuracy $\varepsilon_{\mathrm{est}}$ and measured to return $\widehat{Q}^{\mathrm{rob}}$ (lines 7–8). The only measurement in QRIM is the final extraction of $\widehat{i}$ and $\widehat{Q}^{\mathrm{rob}}(s, a; V)$ for the outer RL loop.

The following proposition formalizes how much QRIM accurately estimates the robust values:

**Proposition 4.3** (QRIM accuracy). *For any $\varepsilon_{\mathrm{cmp}}, \varepsilon_{\mathrm{est}}, \delta \in (0, 1)$, Algorithm 2 returns $\widehat{Q}^{\mathrm{rob}}(s, a; V)$ such that, with probability at least $1 - \delta$,*

$$\big|\widehat{Q}^{\mathrm{rob}}(s, a; V) - Q^{\mathrm{rob}}(s, a; V)\big| \leq \varepsilon_{\mathrm{cmp}} + \varepsilon_{\mathrm{est}}.$$

The proof is provided in Appendix C.2.

*Remark* 4.4. QRIM replaces the classical inner scan with a coherent minimum search. Each candidate one-step return is amplitude-encoded, pairwise comparisons are implemented via QAE at tolerance $\varepsilon_{\mathrm{cmp}}$, and QMF finds the minimizer; a final QAE at $\varepsilon_{\mathrm{est}}$ returns the robust backup. The linear computation over $|\mathcal{U}|$ is reduced to $\mathcal{O}(\sqrt{|\mathcal{U}|})$ queries.

### 4.3. Outer maximization interface

Value-based learners replace classical targets by the robust temporal-difference target $\mathrm{TD}^{\mathrm{rob}}$:

$$\mathrm{TD}^{\mathrm{rob}}(s, a, s') = r(s, a, s') + \gamma \max_{a'} \widehat{Q}^{\mathrm{rob}}(s', a'; V).$$

Policy-gradient learners form a robust critic and advantage

$$\widehat{Q}^{\mathrm{rob}}(s, a; V) \quad \text{and} \quad \widehat{A}^{\mathrm{rob}}(s, a) = \widehat{Q}^{\mathrm{rob}}(s, a; V) - V(s),$$

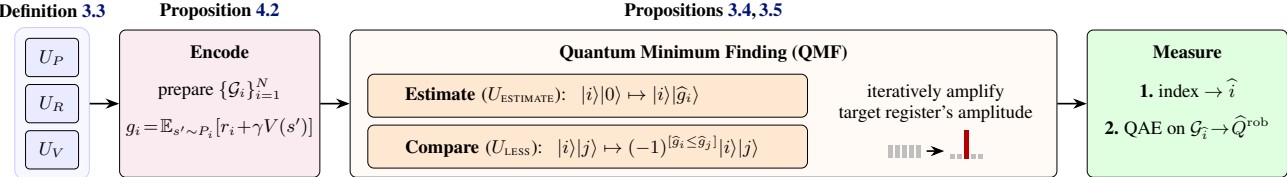

*Figure 2.* **Overview of QRIM.** The three quantum-accessible oracles $(U_P, U_R, U_V)$ compose into $\mathcal{G}_i$, which encodes each candidate's $g_i$ in an ancilla amplitude. *Quantum Minimum Finding* (QMF) iterates *Compare*, which calls *Estimate* (QAE) to read each candidate's amplitude into a value register and phase-flips the smaller; amplitude amplification on the *index* register then boosts the marked candidates in $\mathcal{O}(\sqrt{N})$ queries (vs. $\mathcal{O}(N)$ classical). The Measure block extracts the worst-case index $\widehat{i}$ and re-estimates $\widehat{g}_{\widehat{i}}$ via a final QAE.

and use the usual estimator $\mathbb{E}[\nabla \log \pi_\theta(a|s) \widehat{A}^{\mathrm{rob}}(s,a)]$ with TRPO/PPO-style constraints. Rollout collection and optimization hyperparameters are unchanged. We emphasize that QRIM is compatible with both value-based or policy-gradient RL learners.

QRIM acts pointwise at $(s,a)$ and therefore realizes the rectangular robust operator:

$$(T_{\mathrm{rob}}V)(s) = \max_a Q^{\mathrm{rob}}(s,a;V)$$

For episode-wise robustness at evaluation, we fix the selected index $\widehat{i}$ for an entire trajectory and reuse the same $\mathcal{G}_{\widehat{i}}$; the algorithm itself is unchanged.

*Remark* 4.5. The outer learning loop is unchanged: both value-based and policy-gradient methods simply substitute the robust target $\widehat{Q}^{\mathrm{rob}}$ returned by QRIM. This makes QRIM a drop-in routine that preserves the Max-Min optimization while reducing the inner-loop cost from $\mathcal{O}(N/\varepsilon^2)$ to $\mathcal{O}(\sqrt{N}/\varepsilon)$. Moreover, this estimation error reduces the policy's return by at most $\mathcal{O}(\varepsilon/(1-\gamma)^2)$ for both learners (Appendix C.4), so it does not destabilize the outer training. In essence, QRIM shows how quantum access enables robust RL to be implemented in a more scalable and tractable manner without altering existing training pipelines.

### 4.4. Computational complexity

We count one coherent call of query for the use of $\mathcal{G}_i$ or $\mathcal{G}_i^\dagger$. A classical searching with Monte Carlo uses $\mathcal{O}(N/\varepsilon^2)$ samples to achieve additive error $\varepsilon$ at $(s,a)$. QRIM achieves the quadratic speed-up beyond the classical counterpart, as formalized in the following proposition:

**Proposition 4.6** (Per-call query complexity)**.** *With failure probability $\delta \in (0,1)$, QRIM produces $\widehat{Q}^{\mathrm{rob}}(s,a;V)$ satisfying $|\widehat{Q}^{\mathrm{rob}}(s,a;V) - Q^{\mathrm{rob}}(s,a;V)| \leq \varepsilon_{\mathrm{cmp}} + \varepsilon_{\mathrm{est}}$ using $\tilde{\mathcal{O}}(\sqrt{N}\,\varepsilon_{\mathrm{cmp}}^{-1} \log(1/\delta))$ coherent queries to implement minimum finding, plus $\mathcal{O}(\varepsilon_{\mathrm{est}}^{-1} \log(1/\delta))$ for the final estimation. By choosing $\varepsilon_{\mathrm{cmp}} = \varepsilon_{\mathrm{est}} = \varepsilon/2$, QRIM requires*

$$\tilde{\mathcal{O}}(\sqrt{N}\,\varepsilon^{-1} \log(1/\delta))$$

*total coherent queries, a quadratic improvement over $O(N\varepsilon^{-2})$ of the classical counterpart.*

The proof is provided in Appendix C.5.

A matching lower bound also holds in this setting. Since we do not assume convexity, smoothness, or gradient access with respect to the uncertainty parameter, the inner minimization is an instance of unstructured search over $N$ candidates, and any quantum algorithm solving it requires $\Omega(\sqrt{N})$ queries (Bennett et al., 1997). The bound in Proposition 4.6 matches this lower bound up to a $\log(1/\delta)$ factor (introduced by the success-probability boosting inside QAE), so QRIM is query-optimal in this setting. Comparing Proposition 4.6 to the classical $\mathcal{O}(N/\varepsilon^2)$ Monte Carlo baseline further clarifies the role of the precision parameter $\varepsilon$: QRIM keeps a quadratic improvement in $N$ at every precision level, and additionally reduces the precision dependence from $\mathcal{O}(1/\varepsilon^2)$ to $\mathcal{O}(1/\varepsilon)$.

We note that the bounds above measure the *number of coherent oracle calls* required by QRIM, following the standard convention in quantum algorithm analysis. However, the cost of executing each individual call is a separate matter, governed by the underlying hardware: in the current pre-fault-tolerant era, both classical state-vector emulation and near-term (NISQ) execution (which adds CPU–QPU communication latency) impose substantial per-call overhead, while in a future fault-tolerant regime the per-call cost reduces to the gate depth of the oracle circuit. Appendix D analyzes these three cases, and our experiments report query counts rather than wall-clock time.

## 5. Experiments

We evaluate (i) **robustness** of learned policies and (ii) **inner-loop complexity** of QRIM. Training details, including RL architectures, hyperparameters, and quantum fixed-point precision, are in Appendix G. Under the assumption of access to quantum oracles for classical environment dynamics, we evaluate QRIM in two settings: a tabular domain (FrozenLake-8×8), where the inner loop is implemented with simulated **coherent quantum routines** and the outer learner is a tabular Q-learner; and a continuous-control domain (CartPole Swingup), where the inner loop uses **classical emulation of quantum circuits** and the outer learner

*Table 1.* FrozenLake ranges and grids (implementation defaults). Layouts use 5 holes per case; only connected layouts are admitted. Exact grid values are listed in Appendix G.

| Factor (temporal) | Train set (range; grid) | Evaluation set | |
|---|---|---|---|
| | | Interpolation (range; grid) | Extrapolation (range; grid) |
| Hole layout (Episode-wise) | center $4\times4$ rim; 5 cases | center $4\times4$ rim; 6 cases | outer $6\times6$ rim; 6 cases |
| Slip probability $p$ (Step-wise) | $[0.00, 0.10]$: 4 pts | $[0.00, 0.10]$: 4 pts | $[0.10, 0.25]$: 4 pts |

*Table 2.* CartPole ranges and grids (implementation defaults). Interpolation excludes train points; extrapolation lies strictly outside train ranges with a 15% margin. Train grids imply $N_{\text{step}}=6\times6=36$ damping candidates and $N_{\text{episode}}=5\times5=25$ $(\ell, m)$ candidates. Exact grid values are listed in Appendix G.

| Factor (temporal) | Train set (range; grid) | Evaluation set | |
|---|---|---|---|
| | | Interpolation (range; grid) | Extrapolation (range; grid) |
| Pole length $\ell$ (Episode-wise) | $[0.50, 0.85]$; 5 pts | $[0.50, 0.85]$; 4 pts | $[0.34, 0.50], [0.85, 1.01]$; 4 pts |
| Pole mass $m$ (Episode-wise) | $[0.08, 0.22]$; 5 pts | $[0.08, 0.22]$; 4 pts | $[0.04, 0.08], [0.22, 0.28]$; 4 pts |
| Slider damping $b_{\text{slider}}$ (Step-wise) | $[0.03, 0.12]$; 6 pts | $[0.03, 0.12]$; 4 pts | $[0.02, 0.03], [0.12, 0.18]$; 4 pts |
| Hinge damping $b_{\text{hinge}}$ (Step-wise) | $[0.03, 0.12]$; 6 pts | $[0.03, 0.12]$; 4 pts | $[0.02, 0.03], [0.12, 0.18]$; 4 pts |

is Proximal Policy Optimization (PPO). Also, we provided extended discussions on quantum hardware applicability and report additional experimental results executed on real quantum hardware in Section 6, demonstrating QRIM's robustness under realistic noise conditions. The code is available at `https://github.com/HK-05/QRIM`.

### 5.1. Experiment setup

**Train uncertainty set vs. evaluation perturbation set.** For each environment axis, training uses an axis-aligned range discretized into a grid (the *uncertainty set*). Evaluation uses two disjoint sets per axis: *interpolation* (strictly inside the train range but excluding all train grid points) and *extrapolation* (strictly outside the train range on both sides within physical limits). We also report the nominal performance at the midpoint of the train range. The exact ranges and grid resolutions are summarized in Tables 1 and 2.

**Episode-wise vs. step-wise uncertainty.** Some factors are fixed for an entire episode (episode-wise), while others may change at every step (step-wise). Robust inner minimization is applied at the appropriate temporal level.

**Compared methods.** (i) **Classic RL**: non-robust training at the nominal model; (ii) **Classic RRL**: robust targets via exhaustive scan over the inner grid; (iii) **Quantum RRL** (run via QRIM): robust targets via Grover-style minimum finding (Aer emulation), keeping the outer learner unchanged. The justification of these baseline choices and an empirical comparison with other baselines are provided in Appendix E.

**Protocols and metrics.** We use two protocols: *full-episode-wise* (fixed-per-episode grids with repetitions) and *adversary-episode-wise* (episode-wise sweep while the step-wise adversary selects the worst-case at each step). Let $\phi$ denote the vector of uncertainty factors (e.g., mass, friction).

We report the mean discounted return $\bar{J}$ as the primary metric, averaged over the evaluation uncertainty set $\mathcal{U}_{\text{eval}}$ (i.e., the held-out collection of test environment candidates, disjoint from the training grid). We also visualize per-factor degradation, by evaluating the return as a function of a single factor $\phi_k$ (the $k$-th element of $\phi$), while marginalizing over the remaining factors $\phi_{-k}$. The mean return and the per-factor return curve are defined as:

$$\bar{J} = \frac{1}{|\mathcal{U}_{\text{eval}}|} \sum_{\phi \in \mathcal{U}_{\text{eval}}} J(\phi),$$
$$J_k(\phi_k) = \mathbb{E}_{\phi_{-k} \sim \mathcal{U}_{\text{eval}}^{-k}} \big[ J(\phi_k, \phi_{-k}) \big].$$

### 5.2. Environment details

#### FROZENLAKE-$8\times8$ (TABULAR)

**Axes.** Episode-wise: hole layout (feasible layouts only, ordered by mean $\ell_1$ distance of holes to the center). Step-wise: slip probability $p \in [0, 1]$ (intended action executes with probability of $1 - p$, while left/right directions are taken with probability $p/2$ for each).

**Learner.** Tabular Q-learning variant is adopted for all methods. In robust modes, the worst hole layout is selected once per episode, while step-wise slip probability is adversarially minimized at each transition via the inner routine.

#### CARTPOLE SWINGUP (CONTINUOUS CONTROL)

**Axes.** Episode-wise: pole length $\ell$, mass $m$. Step-wise: slider damping $b_{\text{slider}}$, hinge damping $b_{\text{hinge}}$.

**Learner.** All methods share the same PPO actor-critic, horizons, and optimizer settings. In robust modes, the worst $(\ell, m)$ is selected once per episode (episode-wise), while step-wise damping is adversarially minimized at each transition via the inner routine (scan vs. QRIM).

*Table 3.* Robustness values ($\bar{J}$). Results represent the mean over 6 seeds $\pm$ 95% confidence interval.

| Method | FrozenLake | | | CartPole | | |
|---|---|---|---|---|---|---|
| | Train | Interpolation | Extrapolation | Train | Interpolation | Extrapolation |
| Classic RL | $0.76 \pm 0.17$ | $0.74 \pm 0.15$ | $-0.79 \pm 0.04$ | $61.95 \pm 1.3$ | $60.66 \pm 8.8$ | $26.65 \pm 2.5$ |
| Classic RRL | $0.84 \pm 0.06$ | $0.81 \pm 0.06$ | $-0.45 \pm 0.17$ | $61.03 \pm 0.3$ | $63.24 \pm 2.3$ | $39.86 \pm 15.7$ |
| QRIM (ours) | $\mathbf{0.98 \pm 0.03}$ | $\mathbf{0.98 \pm 0.03}$ | $\mathbf{-0.37 \pm 0.06}$ | $\mathbf{76.41 \pm 1.1}$ | $\mathbf{80.06 \pm 7.9}$ | $\mathbf{58.06 \pm 12.4}$ |

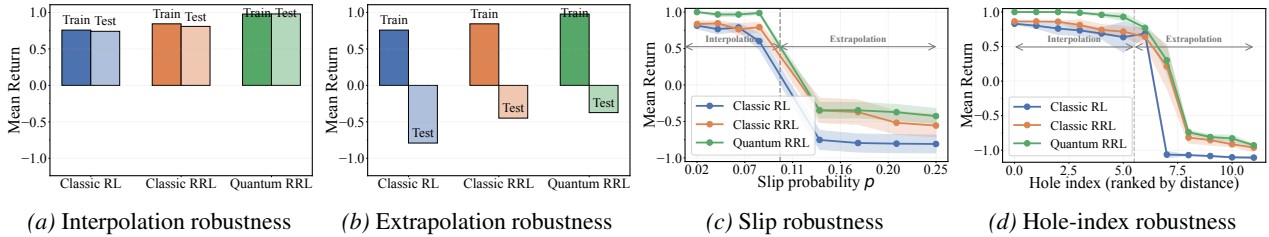

*(a)* Interpolation robustness    *(b)* Extrapolation robustness    *(c)* Slip robustness    *(d)* Hole-index robustness

*Figure 3.* FrozenLake robustness performance and robustness against uncertainty factors.

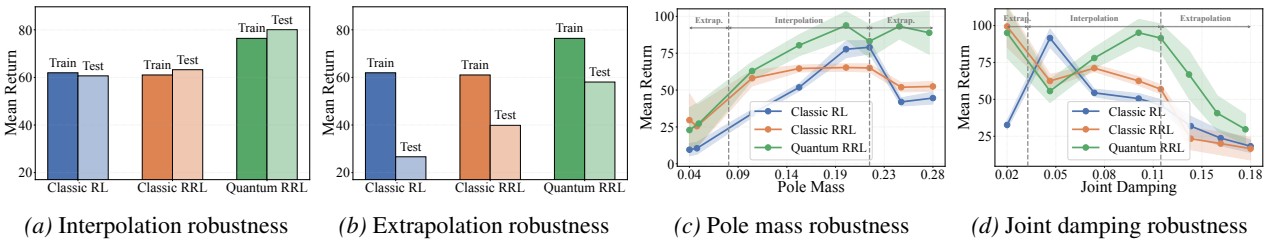

*(a)* Interpolation robustness    *(b)* Extrapolation robustness    *(c)* Pole mass robustness    *(d)* Joint damping robustness

*Figure 4.* CartPole robustness performance and robustness against two factors, i.e., pole mass and joint damping

## 5.3. Robustness

**FrozenLake.** In Figures 3a and 3b, we evaluate $\bar{J}$ between train vs. unseen interpolation and extrapolation sets. In extrapolation, Quantum RRL mitigates performance degradation to $\sim$138% relative to training, significantly outperforming Classic RL, which suffers a $\sim$204% drop, verifying superior stability against out-of-distribution scenarios. As depicted in Figures 3c and 3d, per-factor degradation shows the slip axis and the layout axis of the hole index. The gray dashed line indicates the boundary splitting interpolation and the extrapolation sets. Quantum RRL exhibits substantial robustness, slightly better than the Classic RRL, which confirms its robustness benefits surpass those of non-robust classical RL approaches. Shaded regions of plots indicate the confidence interval. Table 3 reports the numeric performance values.

**CartPole.** In Figures 4a and 4b, Quantum RRL achieves superior average robustness, mitigating degradation to $\sim$24% (against $\sim$57% in Classic RL). As illustrated in Figures 4c and 4d, per-factor robustness isolates sensitivity along $m$ and $b_{\text{joint}}$. $\ell$ and $b_{\text{slider}}$ showed minimal variation and are provided in Appendix G. Shaded regions of plots indicate the confidence interval. The numeric performance values are shown in Table 3.

*Table 4.* Comparison of cumulative oracle calls during training.

| Environment | $N$ | Classical RRL | QRIM |
|---|---|---|---|
| FrozenLake | 4 | 11.45M | **4.11M** (35.9% of Classical RRL) |
| CartPole | 36 | 27.51M | **5.65M** (20.5% of Classical RRL) |

## 5.4. Query complexity

**Protocol.** For classical RRL we count model-evaluation calls across all $N$ candidates. For QRIM, we report *median coherent oracle calls* per robust backup. Unless otherwise stated, tolerance is fixed to $\varepsilon_{\text{cmp}} = \varepsilon_{\text{est}} = 10^{-2}$ and success probability $1 - \delta = 0.9$; the effect of varying $\varepsilon$ on convergence is provided in Appendix F. For the asymptotic scaling analysis (Figures 5a and 5b), we vary the uncertainty grid size $N$. To evaluate the practical benefits during the entire training process (Figures 5c, 5d and Table 4), we use fixed grid sizes and measure the cumulative oracle queries required for convergence.

**Asymptotic query scaling.** We vary the uncertainty grid size $N$ and report the total number of inner-loop queries required for a single robust backup. In FrozenLake, we vary the slip-grid size $N$; the log–log plot shows inner queries versus $N$. QRIM follows a slope $\approx \frac{1}{2}$ while classical scans follow $\approx 1$ (Figure 5a), matching the quadratic improvement. In CartPole, we vary the damping-grid resolution

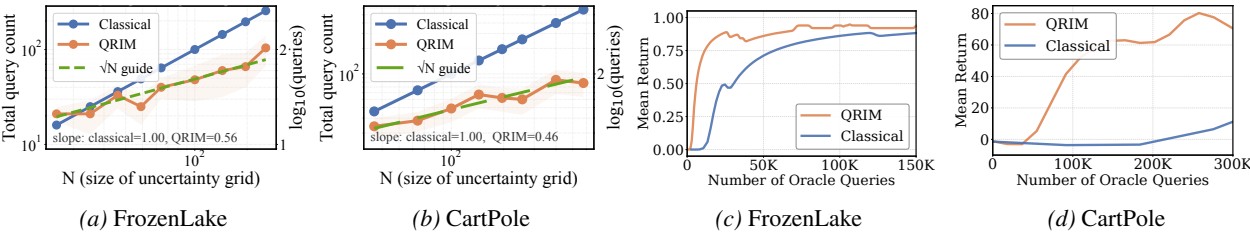

*(a) FrozenLake*      *(b) CartPole*      *(c) FrozenLake*      *(d) CartPole*

*Figure 5.* (a), (b) show the total query counts with respect to the uncertainty size grid $N$ (QRIM slope $\approx \frac{1}{2}$, classical $\approx 1$). (c), (d) show the number of oracle queries required during the training process.

while keeping the episode-wise $(\ell, m)$ grid fixed; it again exhibits the same slope pattern (Figure 5b).

**Query efficiency and convergence.** The advantage in query complexity translates into a substantial reduction in the total cumulative oracle calls throughout the training process. As shown in Table 4, QRIM presents a substantial reduction in query counts compared to classical RRL. Specifically, QRIM only requires 35.9% and 20.5% of the query counts of classical RRL in FrozenLake and CartPole environments, respectively. To demonstrate the speed-up of the training of QRIM, we plot the learning curves in Figures 5c and 5d. QRIM exhibits a significantly faster convergence, along with the number of queries spent. Notably, in the CartPole case, QRIM already reaches almost final performance within 250K queries, whereas classical RRL still struggles to learn a meaningful policy after pouring more than 300K queries.

## 6. Quantum Hardware Verification

To demonstrate the feasibility of our approach on physical quantum devices and validate the noise resilience of QRIM beyond simulations, we conducted experiments using both a realistic noise simulator and real quantum hardware. For a noise model simulation, we utilized the Qiskit Aer noise model, applying depolarizing error rates of 0.05% (1-Qubit error) and 0.6% (2-Qubit error), and a 2% readout error, which were selected to sufficiently reflect the contemporary quantum devices. For real quantum hardware, we used IBM Quantum 127-qubit Eagle processor, with 2 qubits and 20 depth circuits, isolating the quantum search routine while assuming access to quantum oracles for environment dynamics. Due to the limited resource of quantum hardware, experiments were conducted on the FrozenLake-8×8 environment under the extrapolation setting, consistent with the setup described in Section 5.

As shown in Table 5, QRIM executed on real quantum hardware achieved an extrapolation return of -0.52, which is slightly lower than the Classic RRL (-0.45, in Table 3) and the noise-free QRIM (-0.37) due to hardware noise, but significantly outperforms Classic RL (-0.79, in Table 3). The results confirm that QRIM maintains the performance trend

*Table 5.* Comparison of QRIM performance on quantum hardware.

| Metric | FrozenLake QRIM performance | | |
| --- | --- | --- | --- |
| | **noise-free** | **noise model** | **quantum HW** |
| Train | 0.98 | 0.91 | 0.94 |
| Test (extrapolation) | −0.37 | −0.44 | −0.52 |

observed in the noise model simulations (-0.44), demonstrating practical robustness even on noisy intermediate-scale quantum devices (NISQ). Also, the quadratic query speed-up ($\mathcal{O}(\sqrt{|\mathcal{U}|})$) was preserved on real quantum hardware. The total number of queries has been increased only by 5% compared to the noise-free QRIM, which is still much fewer queries than classical RRL.

## 7. Limitation

While we demonstrated feasibility via oracle-based encoding on NISQ hardware, a full quantum implementation for complex environments would require deep arithmetic circuits that exceed current hardware capabilities. Furthermore, the hybrid nature of our setup introduces classical-quantum communication overheads. We anticipate that practical wall-clock speed-ups will be fully realized as quantum hardware matures towards fault-tolerant architectures.

## 8. Conclusion

In this paper, we propose quantum robust inner minimization (QRIM), which demonstrates quadratic speed-up in query complexity beyond its classical counterpart in the robust policy training of RRL. Our approach combines a quantum perspective and the sample scalability issue of robust training for deep models. Specifically, QRIM utilizes a quantum-accessible environment to search the worst-case within the uncertainty set, reducing query complexity from $\mathcal{O}(|\mathcal{U}|)$ to $\mathcal{O}(\sqrt{|\mathcal{U}|})$ via quantum minimum searching. In simulations via classical simulators with a coherent quantum routine and real quantum hardware, we confirm that QRIM trains a robust policy in discrete tabular (FrozenLake) and continuous-control (CartPole) problems with square root reduction of quantum query calls.

## Acknowledgements

This work was supported by the Institute of Information & Communications Technology Planning & Evaluation (IITP) grants funded by the Korea government (Ministry of Science and ICT, MSIT) (No. RS-2020-II201336, Artificial Intelligence Graduate School Program at UNIST), (No. RS-2025-25442824, AI Star Fellowship Program (UNIST)), (No. IITP-2026-RS-2022-00156361, Innovative Human Resource Development for Local Intellectualization Program), and (No. RS-2024-00439803, SW Star Lab). This work was also supported by the National Research Foundation of Korea (NRF) grant funded by the Korea government (MSIT) (No. RS-2024-00459023).

## Impact Statement

This paper introduces a quantum-accelerated algorithm for training robust reinforcement learning agents. By reducing the query complexity of the robust optimization process, our approach also points toward more computationally efficient techniques for handling model uncertainty in reinforcement learning. Given the theoretical nature of this work, we do not foresee immediate negative societal consequences or ethical concerns.

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

## A. Notations for Quantum-Accessible Robust RL

Table 6 summarizes the quantum notation and operators used throughout the paper.

*Table 6.* Quantum-related notation and operators used in our framework.

| Symbol | Meaning |
| --- | --- |
| $\lvert\psi\rangle$ | Quantum state vector in Dirac notation |
| $\lvert s, a\rangle$ | Basis state encoding environment state $s$ and action $a$ |
| $\mathcal{U}$ | Uncertainty set (layouts or slips) |
| $U_P$ | Transition unitary for uncertainty index $i$, encodes $P_i(s'\lvert s, a)$ |
| $U_R$ | Reward oracle/unitary, attaches $r(s, a, s')$ to an ancilla register |
| $U_V$ | Value oracle/unitary, encodes value estimate $V(s')$ |
| $A$ | Amplitude encoding operator (prepares distribution over candidates) |
| $\mathcal{O}$ | Oracle marking "low-reward" candidates in Grover/QRIM |
| $g(s, a; V)$ | One-step backup (Bellman) target |
| $Q^{\mathrm{rob}}(s, a)$ | Robust target $\min_{(P,r)\in\mathcal{U}_{sa}} g(s, a; V)$ |
| $\varepsilon_{\mathrm{cmp}}$ | Comparator tolerance in quantum min-finding |
| $\varepsilon_{\mathrm{est}}$ | Estimation tolerance (accuracy of returned robust value) |

## B. Detailed Construction of Quantum Environment Oracles

This section provides the conceptual circuit-level design of the full quantum oracle unitaries $(U_P, U_R, U_V)$ defined in Definition 3.3. These constructions serve to demonstrate the structural feasibility of our quantum-accessible environment framework and clarify the explicit structure required for a fully coherent quantum oracles. In our experiments, we assume noise-free quantum oracles without explicitly constructing quantum circuits. We present two representative environments that serve complementary roles in demonstrating conceptual feasibility and practical scalability.

### B.1. FrozenLake: Coherent Logic Circuits

For the discrete state space, we construct the full quantum oracles as explicit quantum circuits using standard quantum logic primitives. The process creates a coherent data flow without any classical intervention during the quantum routine.

**Transition Probability Encoding** ($U_P$)   The transition oracle constructs the next-state superposition by combining grid geometry logic with stochastic mixing. The circuit begins with the state $\lvert s\rangle$ and action $\lvert a\rangle$ registers. To encode stochasticity, a multi-controlled $R_y(\theta)$ gate (with $\theta = 2\arcsin(\sqrt{p})$) acts on an auxiliary slip-flag qubit, conditioned on the action register. This creates a superposition of intended ($\lvert 0\rangle_{\mathrm{flag}}$) and slip ($\lvert 1\rangle_{\mathrm{flag}}$) branches. Controlled on these flag states, logical arithmetic gates update the state register $\lvert s\rangle$ to compute the intended next state (e.g., adding the grid width to the index for a Down move) or the slip state. This results in the entangled next-state register $\lvert\psi\rangle_{next} = \sqrt{1-p}\,\lvert s_{\mathrm{intended}}\rangle + \sqrt{p}\,\lvert s_{\mathrm{slip}}\rangle$.

**Reversible Value Loading** ($U_R, U_V$)   The Reward Oracle $U_R$ realizes the reward function via boolean logic. We implement this using a network of Multi-Controlled X (MCX) gates. For instance, if a specific next-state index $s'$ corresponds to a Hole (reward $-1$), an MCX gate targeting the reward register $\lvert r\rangle$ is activated only when the state register matches the binary string of that cell's index. Similarly, the Value Oracle $U_V$ loads the estimated value $V(s')$ via a QROM lookup structure. Finally, a reversible Quantum Adder computes the total return $G = r + \gamma V$, which drives a controlled rotation to encode the amplitude. Crucially, all intermediate compute logic is uncomputed to disentangle the registers.

**Amplitude Encoding**   Finally, to realize QMF for the objective $f(i) = g_i(s, a; V)$ (as defined in Section 3.3), we combine the loaded values. A reversible Quantum Adder first computes the raw return $G = r + \gamma V$. Following Assumption 4.1, this value is normalized to $u = (G - L)/(U - L)$. This scalar $u$ drives a controlled-$R_y(\theta)$ rotation (with $\theta = 2\arcsin(\sqrt{u})$) on a single ancilla qubit. This step strictly implements the state-preparation unitary $\mathcal{G}_i(s, a)$ described in Proposition 4.2, encoding the robust objective into the amplitude $\sqrt{u}\,\lvert 1\rangle$. Crucially, after this encoding, all intermediate compute logic $(U_R, U_V, \mathrm{Adder})$ is uncomputed to disentangle the registers. Figure 6(a) summarizes this circuit at the gate level.

## B.2. CartPole: Classical Emulation of Quantum Oracles

For the CartPole environment, a full circuit implementation requires encoding complex differential equations (e.g., friction, contact dynamics) into reversible quantum arithmetic circuits. As such deep circuits exceed the coherence limits of current hardware and simulators, we employ a classical emulation strategy that mathematically simulates the action of ideal quantum oracles.

**Transition Probability Encoding** ($U_P$)   A rigorous quantum circuit for CartPole dynamics ($U_P$) would necessitate fixed-point quantum registers representing continuous variables ($x, \dot{x}, \theta, \dot{\theta}$) and a sequence of reversible adders, multipliers, and CORDIC circuits to perform time-stepping integration. Since simulating these substantial number of gates is computationally prohibitive and prone to noise on current devices, we instead calculate the resulting quantum amplitudes classically. Our emulation reconstructs the exact quantum state vector that an ideal circuit would produce. In the classical computation step, for a given superposition of state-action pairs $\sum \alpha_i |s_i, a_i\rangle$, the emulator queries the classical physics engine (MuJoCo) to compute the deterministic next state $s'_i = f_{\text{physics}}(s_i, a_i)$ for each basis state $i$, effectively reconstructing the exact quantum state vector.

**Reversible Value Loading** ($U_R, U_V$)   Theoretically, the reward ($U_R$) and value ($U_V$) oracles require arithmetic circuits to evaluate the analytic reward equation and the value network reversibly. In our emulation, we bypass the gate-level construction and directly compute the reward $r_i$ and value estimate $V(s'_i)$ using classic arithmetic based on the state $s'_i$ obtained from the transition step. This process mimics the coherent loading of data into quantum registers $|r, V\rangle$ without the overhead of simulating deep arithmetic logic, ensuring that the values required for the robust objective $g_i(s, a; V)$ are correctly associated with each superposition branch.

**Amplitude Encoding**   Finally, to enable Quantum Amplitude Estimation, the loaded values must be encoded into the amplitude of an ancilla qubit. While a full circuit would employ a reversible Quantum Adder and a controlled rotation, our emulator performs Amplitude Injection. We mathematically calculate the normalized return $u_i = (r_i + \gamma V(s'_i) - L)/(U - L)$ following Assumption 4.1, and directly inject the corresponding amplitude into the state vector: $|\Psi\rangle = \sum_i \alpha_i |s_i, a_i\rangle (\sqrt{u_i} |1\rangle_{\text{anc}} + \dots)$. This constructed state is then passed to the Grover diffusion operator. By counting this batch computation as a single coherent oracle call, we validate the algorithmic query complexity ($\mathcal{O}(\sqrt{N})$) of QRIM, isolating the quantum advantage from hardware constraints.

## B.3. Coherent construction of the LESS oracle

Figure 6(b) shows the gate-level realization of the comparison oracle $U_{\text{LESS}}$ used inside QMF. The construction is fully coherent and contains no intermediate measurement: $U_{\text{ESTIMATE}}$ writes each candidate's QAE estimate $\widehat{g}$ into a value register, a reversible comparator phase-flips the index pair $|i\rangle |j\rangle$ when $\widehat{g}_i \leq \widehat{g}_j$, and $U_{\text{ESTIMATE}}^\dagger$ uncomputes the value registers so that the only register affected at the end is the index pair (with a relative phase).

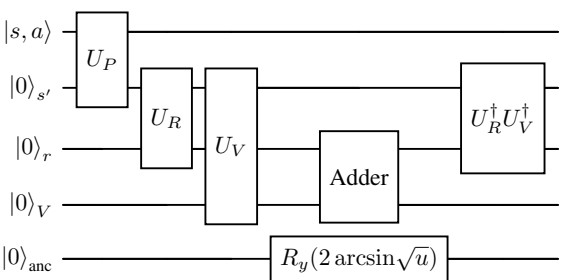

*(a) $\mathcal{G}_i(s, a)$: amplitude encoding of one candidate.*

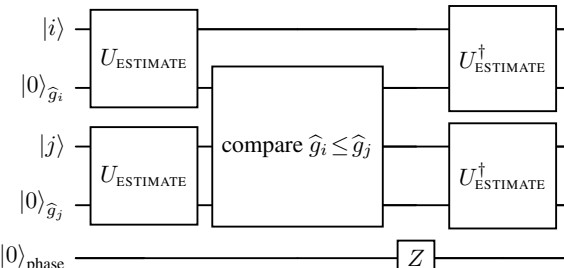

*(b) $U_{\text{LESS}}$: coherent comparator built from two $U_{\text{ESTIMATE}}$ calls (which themselves wrap $\mathcal{G}_i$ via QAE).*

*Figure 6.* **Gate-level construction of QRIM's quantum sub-routines.** (a) $\mathcal{G}_i$ uses $U_P, U_R, U_V$ to load $r, V$ into ancillas, sums them with a reversible Adder, and encodes the normalized return $u = (G - L)/(U - L)$ into an ancilla amplitude via $R_y(2 \arcsin \sqrt{u})$. (b) $U_{\text{LESS}}$ then calls $U_{\text{ESTIMATE}}$ (a coherent QAE wrapping $\mathcal{G}_i$) on each index, runs a reversible comparator + $Z$ for the conditional phase flip, and uncomputes via $U_{\text{ESTIMATE}}^\dagger$.

## C. Proofs

### C.1. Amplitude encoding of $g_i$ (Proposition 4.2)

*Proof.* Under Assumption 4.1, $x := r_i(s, a, s') + \gamma V(s') \in [L, U]$. Define the normalized quantity

$$u(s') := \frac{x - L}{U - L} \in [0, 1]. \tag{4}$$

Consider registers $|s, a\rangle |0_{S'}\rangle |0_{\mathrm{wrk}}\rangle |0_{\mathrm{anc}}\rangle$. Apply the transition sampler for candidate $i$:

$$U_P : |s, a, 0\rangle \mapsto \sum_{s' \in \mathcal{S}} \sqrt{P_i(s'|s, a)} |s, a, s'\rangle. \tag{5}$$

Load reward and value to a work register via reversible LUTs:

$$U_R : |s, a, s', 0\rangle \mapsto |s, a, s', r_i(s, a, s')\rangle, \tag{6}$$
$$U_V : |s', 0\rangle \mapsto |s', V(s')\rangle. \tag{7}$$

Compute $x = r_i(s, a, s') + \gamma V(s')$ in fixed point, then $u = (x - L)/(U - L)$ via a reversible divider. Perform a controlled $R_y$ on the ancilla with angle $2 \arcsin \sqrt{u}$:

$$|u\rangle |0_{\mathrm{anc}}\rangle \mapsto |u\rangle \left( \sqrt{u} |1\rangle + \sqrt{1 - u} |0\rangle \right). \tag{8}$$

Uncompute all work registers. The overall state becomes

$$\sum_{s'} \sqrt{P_i(s'|s, a)} |s, a, s'\rangle \left( \sqrt{u(s')} |1\rangle + \sqrt{1 - u(s')} |0\rangle \right). \tag{9}$$

Let $\mu_i := \mathbb{E}_{s' \sim P_i}[u(s')]$. By linearity of expectation, amplitude estimation on the ancilla yields $\widehat{\mu}_i$ with $|\widehat{\mu}_i - \mu_i| \leq \varepsilon$ using $\mathcal{O}(\varepsilon^{-1} \log(1/\delta))$ coherent queries. Since

$$g_i(s, a; V) = \mathbb{E}_{s' \sim P_i}[x] = L + (U - L) \mu_i, \tag{10}$$

we output $\widehat{g}_i := L + (U - L) \widehat{\mu}_i$. $\qquad \square$

### C.2. QRIM accuracy (Proposition 4.3)

*Proof.* Let the true ordered values be $g_{(1)} \leq g_{(2)} \leq \cdots \leq g_{(N)}$. Each comparator call $U_{\mathrm{LESS}}(i, j)$ returns the phase $(-1)^{[\widehat{g}_i \leq \widehat{g}_j]}$ where

$$|\widehat{g}_i - g_i| \leq \varepsilon_{\mathrm{cmp}}, \quad |\widehat{g}_j - g_j| \leq \varepsilon_{\mathrm{cmp}} \tag{11}$$

hold simultaneously with probability $\geq 1 - \delta_{\mathrm{cmp}}$ by QAE. Thus, whenever $g_i \leq g_j - 2\varepsilon_{\mathrm{cmp}}$ the comparator returns the correct order. Equivalently, the comparator may err only on *near ties*, i.e.,

$$|g_i - g_j| \leq 2\varepsilon_{\mathrm{cmp}}. \tag{12}$$

QMF with a noisy comparator of error rate $\eta < 1/2$ returns some $\widehat{i}$ satisfying (e.g., noisy-oracle analyses (Durr & Hoyer, 1996))

$$g_{\widehat{i}} \leq g_{(1)} + 2\varepsilon_{\mathrm{cmp}} \tag{13}$$

with probability at least $1 - \delta/2$ after polylogarithmic repetitions (absorbed into $\widetilde{\mathcal{O}}(\cdot)$). Finally, we re-estimate $g_{\widehat{i}}$ to accuracy $\varepsilon_{\mathrm{est}}$:

$$|\widehat{g}_{\widehat{i}} - g_{\widehat{i}}| \leq \varepsilon_{\mathrm{est}} \tag{14}$$

with probability at least $1 - \delta/2$. By a union bound and (13)–(14),

$$\left| \widehat{g}_{\widehat{i}} - g_{(1)} \right| \leq \underbrace{\left| \widehat{g}_{\widehat{i}} - g_{\widehat{i}} \right|}_{\leq \varepsilon_{\mathrm{est}}} + \underbrace{\left| g_{\widehat{i}} - g_{(1)} \right|}_{\leq 2\varepsilon_{\mathrm{cmp}}} \leq 2\varepsilon_{\mathrm{cmp}} + \varepsilon_{\mathrm{est}} \tag{15}$$

with probability at least $1 - \delta$. Replacing the factor 2 by 1 is possible by setting the decision threshold in $U_{\mathrm{LESS}}(i, j)$ at $\varepsilon_{\mathrm{cmp}}$-margins; we keep the simpler form and state the bound as $\varepsilon_{\mathrm{cmp}} + \varepsilon_{\mathrm{est}}$ up to a constant factor absorbed by redefining $\varepsilon_{\mathrm{cmp}}$ (customarily done in QAE analyses). $\qquad \square$

## C.3. Approximate robust operator and fixed-point stability

Let $\widehat{Q}^{\mathrm{rob}}(s, a; V)$ satisfy the uniform bound

$$\left| \widehat{Q}^{\mathrm{rob}}(s, a; V) - Q^{\mathrm{rob}}(s, a; V) \right| \leq \epsilon, \qquad \forall (s, a), \; \forall V, \tag{16}$$

with $\epsilon := \varepsilon_{\mathrm{cmp}} + \varepsilon_{\mathrm{est}} + c\eta_{\mathrm{arith}}$. Define

$$(\widehat{T}_{\mathrm{rob}} V)(s) := \max_a \widehat{Q}^{\mathrm{rob}}(s, a; V). \tag{17}$$

**Lemma C.1** (Operator deviation). *For any $V$, $\|(\widehat{T}_{\mathrm{rob}} - T_{\mathrm{rob}})V\|_\infty \leq \epsilon$.*

*Proof.* For each $s$,

$$\left| (\widehat{T}_{\mathrm{rob}} V)(s) - (T_{\mathrm{rob}} V)(s) \right| = \left| \max_a \widehat{Q}^{\mathrm{rob}}(s, a; V) - \max_a Q^{\mathrm{rob}}(s, a; V) \right| \tag{18}$$

$$\leq \max_a \left| \widehat{Q}^{\mathrm{rob}}(s, a; V) - Q^{\mathrm{rob}}(s, a; V) \right| \leq \epsilon. \tag{19}$$

$\square$

**Theorem C.2** (Fixed-point perturbation). *Let $V^*$ be the unique fixed point of $T_{\mathrm{rob}}$ and $\widehat{V}$ be the unique fixed point of $\widehat{T}_{\mathrm{rob}}$. Then*

$$\| \widehat{V} - V^* \|_\infty \leq \frac{\epsilon}{1 - \gamma}. \tag{20}$$

*Proof.* By $\gamma$-contraction of $T_{\mathrm{rob}}$ and $\widehat{T}_{\mathrm{rob}}$ and Lemma C.1,

$$\| \widehat{V} - V^* \|_\infty = \| \widehat{T}_{\mathrm{rob}} \widehat{V} - T_{\mathrm{rob}} V^* \|_\infty \tag{21}$$

$$\leq \| \widehat{T}_{\mathrm{rob}} \widehat{V} - T_{\mathrm{rob}} \widehat{V} \|_\infty + \| T_{\mathrm{rob}} \widehat{V} - T_{\mathrm{rob}} V^* \|_\infty \tag{22}$$

$$\leq \epsilon + \gamma \| \widehat{V} - V^* \|_\infty. \tag{23}$$

Rearrange to obtain (20). $\square$

## C.4. Outer-loop policy suboptimality (Remark 4.5)

Theorem C.2 bounds the perturbation of the inner robust value function under QRIM error. The following result extends this bound to the policy actually executed by the outer learner, covering both value-based and policy-gradient instantiations of QRIM.

**Theorem C.3** (Outer-loop policy suboptimality). *Suppose $|\widehat{Q}^{\mathrm{rob}}(s, a; V) - Q^{\mathrm{rob}}(s, a; V)| \leq \epsilon$ for all $(s, a)$ and all $V$, and let $V^*, \widehat{V}$ be the fixed points of $T_{\mathrm{rob}}$ and $\widehat{T}_{\mathrm{rob}}$, respectively, so that $\|\widehat{V} - V^*\|_\infty \leq \epsilon/(1 - \gamma)$ by Theorem C.2. Then:*

*(i) (**Value-based.**) Any deterministic policy $\widehat{\pi}$ greedy with respect to $\widehat{Q}^{\mathrm{rob}}(\cdot, \cdot; \widehat{V})$ satisfies, for every state $s$,*

$$V^{\widehat{\pi}}(s) \geq V^*(s) - \frac{2\gamma\epsilon}{(1 - \gamma)^2}. \tag{24}$$

*(ii) (**Policy-gradient.**) For an actor–critic learner that uses $\widehat{Q}^{\mathrm{rob}}$ as the critic and updates the policy under a KL trust region of radius $\delta_{\mathrm{KL}}$ (e.g., TRPO/PPO (Schulman et al., 2015)), the converged policy $\widehat{\pi}$ satisfies*

$$V^{\widehat{\pi}}(s) \geq V^*(s) - \frac{C\epsilon}{(1 - \gamma)^2}, \tag{25}$$

*where $C$ is a constant depending on $\delta_{\mathrm{KL}}$ and the maximum advantage range, but not on $N$.*

*Proof.* **Part (i).** By Theorem C.2,

$$\|\widehat{V} - V^*\|_\infty \ \le \ \frac{\epsilon}{1 - \gamma}. \tag{26}$$

The standard greedy-policy loss bound (Singh & Yee, 1994) applied to $\widehat{V}$ states that, for any policy $\widehat{\pi}$ greedy with respect to $\widehat{V}$,

$$V^{\widehat{\pi}}(s) \ \ge \ V^*(s) \ - \ \frac{2\gamma}{1 - \gamma} \|\widehat{V} - V^*\|_\infty \quad \text{for all } s. \tag{27}$$

Substituting (26) into (27) yields

$$V^{\widehat{\pi}}(s) \ \ge \ V^*(s) \ - \ \frac{2\gamma\epsilon}{(1 - \gamma)^2} \quad \text{for all } s, \tag{28}$$

which is the claimed bound.

**Part (ii).** Writing $A^{\mathrm{rob}}(s,a) = Q^{\mathrm{rob}}(s,a;V^*) - V^*(s)$ and the same decomposition for $\widehat{A}^{\mathrm{rob}}$, the per-state advantage error satisfies

$$|\widehat{A}^{\mathrm{rob}}(s,a) - A^{\mathrm{rob}}(s,a)| \le |\widehat{Q}^{\mathrm{rob}}(s,a;\widehat{V}) - Q^{\mathrm{rob}}(s,a;V^*)| \ + \ |\widehat{V}(s) - V^*(s)| \tag{29}$$

$$\le \epsilon \ + \ \|\widehat{V} - V^*\|_\infty \tag{30}$$

$$\le \frac{2\epsilon}{1 - \gamma}, \tag{31}$$

where the last step uses (26) together with $\epsilon \le \epsilon/(1 - \gamma)$.

We make the standard policy-gradient assumption that the actor uses trust-region updates with a fixed KL radius $\delta_{\mathrm{KL}}$ at every step and that the resulting sequence of policies converges to some policy $\widehat{\pi}$. Under this assumption, the per-step improvement bound of Schulman et al. (2015) controls the suboptimality of each update by a multiple of the $\ell_\infty$ advantage error and $\sqrt{\delta_{\mathrm{KL}}}$. Substituting (31) into this per-step bound and propagating it to the fixed point gives

$$V^{\widehat{\pi}}(s) \ \ge \ V^*(s) \ - \ \frac{C\,\epsilon}{(1 - \gamma)^2} \quad \text{for all } s, \tag{32}$$

where $C$ depends on $\delta_{\mathrm{KL}}$ and the maximum advantage range, but not on $N$. $\qquad\square$

### C.5. Per-call query complexity (Proposition 4.6)

*Proof.* Throughout this section, we use $M_{\mathrm{alg}}(\cdot)$ to denote the *query complexity* (i.e., the number of coherent oracle calls) required by algorithm "alg" at the specified accuracy and confidence parameters. Here $\mathcal{G}_i$ denotes the amplitude-preparation unitary for candidate $i$, which maps the all-zero state to the superposition in (9), and $\mathcal{G}_i^\dagger$ is its inverse. A single QAE call achieving additive tolerance $\varepsilon$ with failure $\le \delta'$ uses

$$M_{\mathrm{QAE}}(\varepsilon, \delta') \ = \ \mathcal{O}\big(\varepsilon^{-1} \log(1/\delta')\big) \tag{33}$$

coherent queries to $\mathcal{G}_i$ and $\mathcal{G}_i^\dagger$ (Brassard et al., 2000). The QMF over $N$ items with access to a *mark* oracle (a comparator phase) has complexity

$$M_{\mathrm{QMF}}(N, \delta/2) \ = \ \tilde{\mathcal{O}}\big(\sqrt{N} \log(1/\delta)\big). \tag{34}$$

Implementing one comparator requires *two* QAE calls at accuracy $\varepsilon_{\mathrm{cmp}}$; hence the search phase costs

$$M_{\mathrm{search}} \ = \ \tilde{\mathcal{O}}\big(\sqrt{N} \cdot M_{\mathrm{QAE}}(\varepsilon_{\mathrm{cmp}}, \delta')\big) \ = \ \tilde{\mathcal{O}}\big(\sqrt{N}\,\varepsilon_{\mathrm{cmp}}^{-1} \log(1/\delta)\big), \tag{35}$$

where we set $\delta' = \Theta(\delta/\log N)$ and absorb polylog factors. The final estimate contributes

$$M_{\mathrm{final}} \ = \ \mathcal{O}\big(\varepsilon_{\mathrm{est}}^{-1} \log(1/\delta)\big). \tag{36}$$

Choosing $\varepsilon_{\mathrm{cmp}} = \varepsilon_{\mathrm{est}} = \varepsilon/2$ yields the total

$$M_{\mathrm{QRIM}} \ = \ \tilde{\mathcal{O}}\big(\sqrt{N}\,\varepsilon^{-1} \log(1/\delta)\big). \tag{37}$$

A classical exhaustive scan that estimates every $g_i$ by Monte Carlo to accuracy $\varepsilon$ requires

$$M_{\mathrm{classical}} \ = \ \Theta\big(N\,\varepsilon^{-2} \log(1/\delta)\big) \tag{38}$$

samples, giving the quadratic improvement simultaneously in $N$ and $1/\varepsilon$. $\qquad\square$

## C.6. Extension to continuous uncertainty sets

Our main analysis (Section 3.2) treats $\mathcal{U}_{sa}$ as a given finite candidate set of size $N$, which is the standard formulation of robust RL with discrete uncertainty (Nilim & Ghaoui, 2005; Rajeswaran et al., 2017). Here we describe how the analysis extends when the candidates $\{(P_i, r_i)\}_{i=1}^N$ are obtained by discretizing an underlying continuous uncertainty set $\mathcal{U}_{sa}^{\text{cont}}$ parameterized by $u \in \mathcal{U}_{sa}^{\text{cont}}$. Let $g(s, a, u; V) := \mathbb{E}_{s' \sim P_u(\cdot|s,a)}[r_u(s, a, s') + \gamma V(s')]$ denote the one-step return as a function of the continuous parameter $u$, and assume $u \mapsto g(s, a, u; V)$ is Lipschitz with constant $L$ under a chosen metric on $\mathcal{U}_{sa}^{\text{cont}}$. For a discretization grid $\{u_1, \ldots, u_N\}$ with mesh size $\Delta$, the induced uniform discretization error is

$$\varepsilon_{\text{disc}} := \sup_{(s,a)} \big| \min_{u \in \mathcal{U}_{sa}^{\text{cont}}} g(s, a, u; V) - \min_{i \in [N]} g(s, a, u_i; V) \big| \leq L\Delta.$$

Let $V^*$ denote the fixed point of the continuous-case robust operator and $\widehat{V}^*$ that of its discretized counterpart. By the $\gamma$-contraction of $T_{\text{rob}}$ in $\|\cdot\|_\infty$ (Proposition 3.2),

$$\|V^* - \widehat{V}^*\|_\infty \leq \frac{\varepsilon_{\text{disc}}}{1 - \gamma}.$$

Combined with the QRIM estimation error of Proposition 4.3 ($\varepsilon_{\text{cmp}} + \varepsilon_{\text{est}}$), the total end-to-end deviation of the executed value function is bounded by

$$\|V^* - \widehat{V}\|_\infty \leq \frac{\varepsilon_{\text{disc}} + \varepsilon_{\text{cmp}} + \varepsilon_{\text{est}}}{1 - \gamma}.$$

where $\widehat{V}$ is the fixed point of the approximate operator $\widehat{T}_{\text{rob}}$ defined in Appendix C.3. Thus, when a continuous uncertainty set admits a Lipschitz parameterization, QRIM's discrete formulation incurs only a controlled additive error, and the quadratic query advantage in $N$ is preserved.

# D. Per-call Computational Overhead Across Hardware Regimes

The complexity bounds in Section 4.4 count coherent oracle queries, the standard convention in quantum algorithm analysis. Translating these bounds into wall-clock cost requires accounting for the cost of executing each individual call, which depends entirely on the underlying hardware. We discuss the three relevant settings below.

**Classical state-vector emulation.** On a classical machine, simulating an oracle that acts on $q$ qubits requires storing and updating a state vector of dimension $2^q$, so each gate incurs $\mathcal{O}(2^q)$ floating-point operations. The total cost of a single oracle call therefore scales exponentially in the qubit count, and the $\mathcal{O}(\sqrt{N})$ reduction in query count does not translate into any computational advantage in this regime. Emulation remains useful for validating algorithmic scaling on small problem instances—including all simulator-based experiments in this paper—but it cannot be used to claim a wall-clock speedup.

**Near-term (NISQ) execution.** On current quantum hardware, the per-call cost is dominated by two factors: the gate depth of the oracle circuit and the CPU–QPU communication latency incurred at every call. For tabular environments with a shallow oracle circuit (e.g., the FrozenLake oracle in Appendix B), reliable NISQ execution is already feasible, and we report real-hardware results from an IBM 127-qubit device in Section 6. For continuous-control environments, the oracle requires reversible arithmetic over fixed-precision registers, with $\mathcal{O}(b^2)$ gates per arithmetic operation at $b$-bit precision. As an estimate for our CartPole environment at $b = 8$, the state register uses $4 \times 8 = 32$ qubits, with roughly 200 ancilla qubits for intermediate computations, totaling about 200–250 qubits and 2,000–3,000 gates per oracle call. Under representative NISQ two-qubit gate error rates of $0.5\%$, the 2,000-gate circuit fidelity is $(0.995)^{2000} \approx 4.4 \times 10^{-5}$, well below any threshold required for meaningful execution. Continuous-control oracles of this depth are therefore not yet realizable on NISQ hardware. As $N$ grows, however, the $\mathcal{O}(\sqrt{N})$ query reduction would dominate any constant-factor latency overhead even in the NISQ regime, provided the oracle can be reliably executed.

**Fault-tolerant execution.** Under error-corrected quantum hardware, the per-call cost reduces to the gate depth of the oracle circuit, eliminating both the exponential overhead of classical emulation and the noise constraints of NISQ devices. In this regime, the $\mathcal{O}(\sqrt{N})$ query reduction translates directly into a computational advantage. Comparing the classical per-step cost $\mathcal{O}(N)$ to the quantum per-step cost $\mathcal{O}(\sqrt{N} \cdot b^2)$ yields a wall-clock crossover at $N > b^4$; for $b = 8$, this corresponds to $N > 4{,}096$ candidate environments, beyond which QRIM is unconditionally faster than any classical method for the inner minimization.

## E. Discussion of Baseline Choices and Other Baseline

A wide range of classical and quantum algorithms have been developed for minimax problems—e.g., zeroth-order min-max methods (ZO-MIN-MAX), gradient-free saddle-point methods, and quantum matrix-game / saddle-point solvers. These all require structural conditions on the inner objective such as smoothness, convexity, or continuous parameterization with gradient access, none of which our setting provides: the inner minimization operates over a finite discrete candidate set $\mathcal{U}_{sa}$, with each $g_i$ accessible only through evaluation. Discrete-set methods that respect this access pattern, most notably best-arm identification (BAI), do apply in principle, but they reduce only constant factors and remain $\mathcal{O}(N)$ in $|\mathcal{U}_{sa}|$; they cannot achieve sub-linear scaling. Combined with the $\Omega(N)$ classical and $\Omega(\sqrt{N})$ quantum lower bounds discussed in Section 4.4, QRIM is the first method to attain optimal sub-linear scaling for the inner minimization in this setting.

To validate this empirically, we additionally compare QRIM against a best-arm-identification baseline (BAI-RRL). BAI adaptively allocates samples to identify the worst-case candidate with high-probability guarantees and is the strongest classical baseline that respects the discrete, evaluation-only access pattern of our setting (no convexity, smoothness, or gradient access). We instantiate BAI-RRL on FrozenLake under the same training protocol as Classical RRL and QRIM (Sec. 5), and report cumulative oracle calls together with robustness on the train, interpolation, and extrapolation sets.

*Table 7.* Comparison with the BAI-RRL baseline on FrozenLake. We report cumulative oracle calls during training (*Query complexity*; percentages relative to Classical RRL) and robustness $\bar{J}$ (*Robustness*; mean over 6 seeds $\pm$ 95% confidence interval) on train, interpolation, and extrapolation sets. Classic RL is included for reference; it does not perform inner minimization, so its inner-loop query count is not directly comparable (marked —). QRIM attains the largest query reduction while improving robustness over all baselines.

| Method | Query complexity | Robustness | | |
|---|---|---|---|---|
| | Queries | Train | Interpolation | Extrapolation |
| Classic RL | — | $0.76 \pm 0.17$ | $0.74 \pm 0.15$ | $-0.79 \pm 0.04$ |
| Classical RRL | 11.45M | $0.84 \pm 0.06$ | $0.81 \pm 0.06$ | $-0.45 \pm 0.17$ |
| BAI-RRL | 9.67M (84.5%) | $0.96 \pm 0.02$ | $0.95 \pm 0.02$ | $-0.49 \pm 0.07$ |
| **QRIM (ours)** | **4.11M (35.9%)** | $\mathbf{0.98 \pm 0.03}$ | $\mathbf{0.98 \pm 0.03}$ | $\mathbf{-0.37 \pm 0.06}$ |

As shown in Table 7, all robust methods clearly outperform Classic RL—most notably on extrapolation, where Classic RL collapses to $-0.79$ while every robust variant remains above $-0.50$. BAI-RRL reduces cumulative oracle calls by 15.5% relative to Classical RRL and substantially improves train and interpolation robustness, but its extrapolation return is slightly worse than Classical RRL. QRIM, in contrast, reduces queries by 64.1% and attains the best robustness across all evaluation splits. This is consistent with the scaling argument above: BAI improves constant factors but remains $\mathcal{O}(N)$, whereas QRIM is the only method achieving the optimal $\mathcal{O}(\sqrt{N})$ scaling. We restrict BAI-RRL to FrozenLake because the comparison is most informative in the tabular setting where queries are directly counted at the candidate level; the asymptotic conclusion carries over to CartPole.

## F. Convergence under varying QAE precision $\varepsilon$

To examine how QRIM's stochastic estimation error propagates into PPO training, we run an ablation on Cart-Pole in which the QAE comparison and estimation tolerances are coupled, $\varepsilon_{\mathrm{cmp}} = \varepsilon_{\mathrm{est}} = \varepsilon$, and varied across $\varepsilon \in \{0.01, 0.05, 0.10, 0.15, 0.20\}$. The setting $\varepsilon = 0.01$ corresponds to the value used in our main experiments (Sec 5). All other hyperparameters and the outer PPO loop are held fixed; results are mean evaluation returns over 6 seeds.

Training is essentially unaffected up to $\varepsilon = 0.10$ (5% degradation in train return), and evaluation robustness on the extrapolation set degrades only mildly in this range. Beyond $\varepsilon = 0.15$, performance drops more sharply (19% on train, 34% on extrapolation), indicating the practical tolerance boundary for the outer PPO loop. The ablation thus establishes a usable precision margin for QRIM in policy-gradient training.

*Table 8.* Convergence of PPO under QRIM with varying QAE precision $\varepsilon$ (CartPole). Mean evaluation return over 6 seeds; $\varepsilon = 0.01$ is the value used in the main experiments.

| Metric | $\varepsilon$=0.01 | $\varepsilon$=0.05 | $\varepsilon$=0.10 | $\varepsilon$=0.15 | $\varepsilon$=0.20 |
|---|---|---|---|---|---|
| Train | 76.41 | 77.27 | 72.96 | 62.14 | 56.14 |
| Extrapolation | 58.06 | 54.34 | 52.05 | 38.23 | 27.04 |

## G. Experiments details

**FrozenLake (tabular, value-based).** We use the $8\times8$ map with start $(0,0)$ and goal $(7,7)$. Episode-wise uncertainty is given by hole layouts sampled from the central $4\times4$ rim, with $H{=}5$ holes per layout. Five layouts are used for training; interpolation layouts come from the same rim but disjoint from training; extrapolation layouts come from the $6\times6$ rim. Step-wise uncertainty is the slip probability $p$, with training grid $\{0.000, 0.033, 0.067, 0.100\}$; interpolation uses midpoints inside $(0.000, 0.100)$; extrapolation extends up to 0.25. Tabular Q-learning is used with $\gamma{=}0.99$, learning rate $\alpha{=}0.1$, and $\epsilon$-greedy exploration decayed linearly from 0.10 to 0.02 over the first 30% of episodes. Exploring starts are used with probability 0.35. Robust variants blend worst-case and mean backups with factor 0.7, and select the current worst layout with probability 0.75.

**CartPole swingup (dm_control, policy-gradient).** Episode-wise uncertainty is the pole length $\ell$ and mass $m$; step-wise uncertainty is slider damping $b_{\text{slider}}$ and joint damping $b_{\text{joint}}$. Training grids: $5\times5$ for $(\ell, m)$ and $6\times6$ for $(b_{\text{slider}}, b_{\text{joint}})$; interpolation excludes training points; extrapolation expands ranges by 15%. Concretely, the equis-paced training grids are $\ell \in \{0.500, 0.5875, 0.675, 0.7625, 0.850\}$, $m \in \{0.080, 0.115, 0.150, 0.185, 0.220\}$, and $b_{\text{slider}}, b_{\text{joint}} \in \{0.030, 0.048, 0.066, 0.084, 0.102, 0.120\}$. Interpolation grids use 4 midpoints inside each training interval (with 0.5-fraction offset); extrapolation grids use 4 points equispaced on the two outer extension intervals listed in Table 2. We train PPO with identical hyperparameters across baselines: two-layer MLP (128 units, Tanh) for actor and critic, clip ratio 0.2, learning rates $3 \times 10^{-4}$ (policy) and $10^{-3}$ (value), GAE $\lambda{=}0.95$, horizon 2048, minibatch size 256, and 40 SGD updates per epoch. Discount $\gamma{=}0.99$. Robust RRL scans all damping candidates per step; QRIM calls Grover min-finding on the damping grid while leaving the PPO pipeline unchanged. Episode-wise robust choice of $(\ell, m)$ is made using the critic at the start of each rollout.

**CartPole per-factor degradation against pole length and slider damping.** Fig. 8 shows the robustness of each algorithm evaluated against variations in pole length and slider damping.

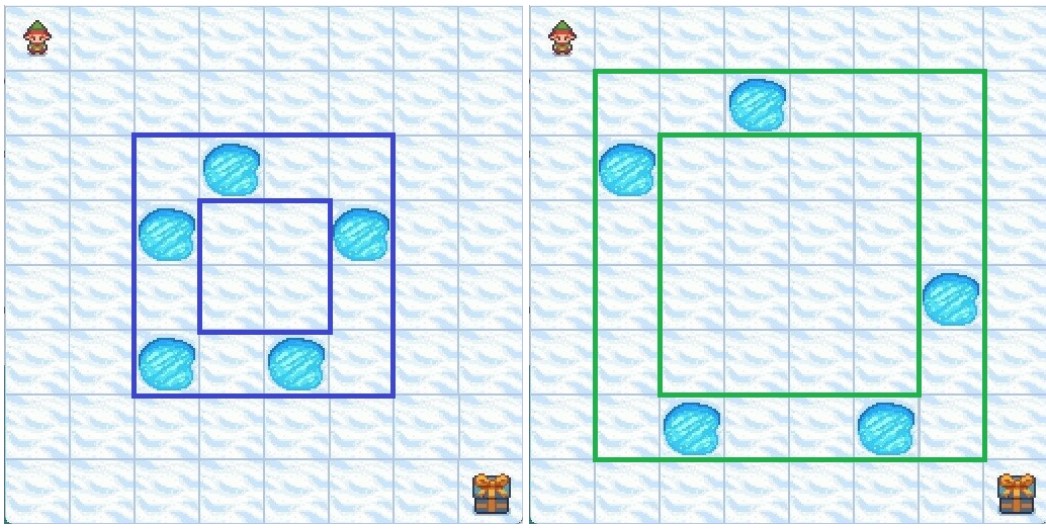

*(a)* Train layouts ($4\times4$ rim).          *(b)* Test layouts (extrapolation).

*Figure 7.* **FrozenLake uncertainty sets.** Left: training layouts sampled on the central $4\times4$ rim. Right: evaluation layouts for extrapolation (central $6\times6$ rim).

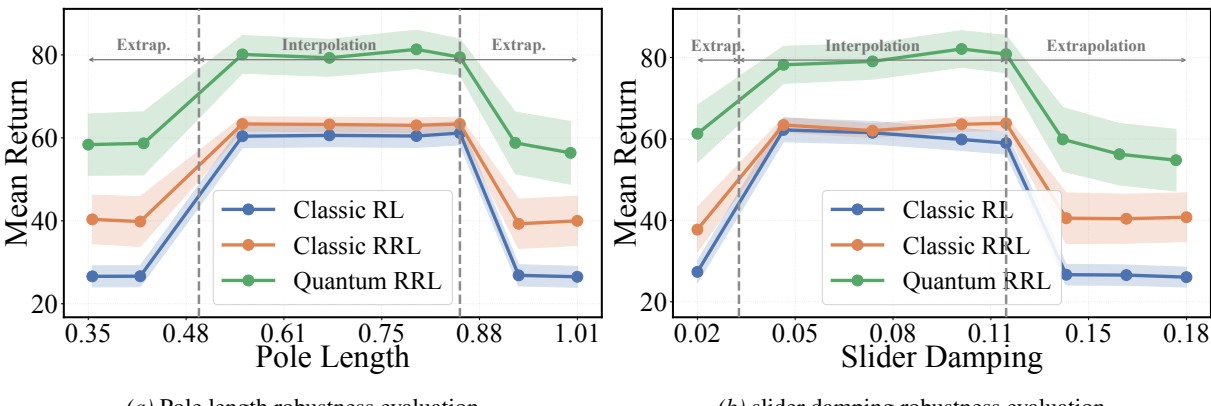

*(a)* Pole length robustness evaluation          *(b)* slider damping robustness evaluation

*Figure 8.* CartPole robustness against additional factors, i.e., pole length and slider damping

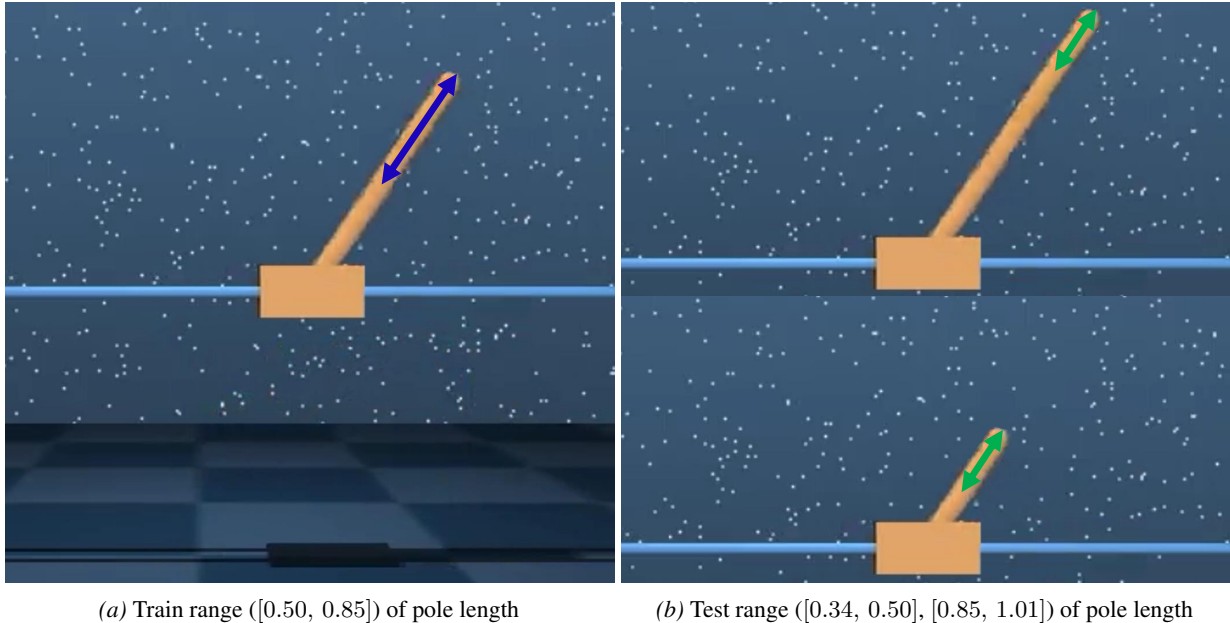

*(a)* Train range ($[0.50, 0.85]$) of pole length          *(b)* Test range ($[0.34, 0.50]$, $[0.85, 1.01]$) of pole length

*Figure 9.* **CartPole uncertainty sets.** Train (Left) and Test (Right) range of the pole length.

## H. Discussions on QRIM's implementation on Quantum Hardware

Our theory provides an intuition about its practicality in noisy quantum hardware. In the scale behavior of QRIM, i.e., $\tilde{\mathcal{O}}(\sqrt{N}\log(1/\delta))$ (referring to Proposition 3.5), $\delta$ means the failure probability for minimum search. In our noise-free simulations, we already set an imperfect search with a success probability of $(1-\delta) = 0.90 < 1$. It reflects the stochastic behavior of quantum environments. In the related literature, hardware noise is shown to degrade the success probability, which is another factor that might affect the success of minimum search.

As we already assume imperfect minimum search, i.e., $(1-\delta) = 0.90$, in our experiments, QRIM's successful training even with such imperfect search makes us conjecture QRIM's possible robustness against hardware noise, which also disturbs perfect search.

This leads us to conjecture that QRIM shows sufficient robustness in both noise-model-based experiments and real quantum hardware, coinciding with the previous noise-free results. Also, if we set a severe failure probability (a low $\delta$) for QRIM, it may lead to a slight increase in the required query counts to keep the robustness performance.

