# OpenReview forum: "Quantum Robust Inner Minimization for Reinforcement Learning with Quadratic Speed-Up in Query Complexity"
_ICML.cc/2026/Conference — ICML 2026 regular_

### Official Review · Reviewer_fvMD · 2026-03-03

**Soundness:** 3
**Presentation:** 2
**Significance:** 3
**Originality:** 2
**Overall Recommendation:** 3
**Confidence:** 5

**Summary:**

This paper proposes a quantum robust inner minimization (QRIM) algorithm for robust reinforcement learning. The key idea is to encode environment transition and reward information into quantum-accessible oracles and then apply existing quantum subroutines, notably quantum amplitude estimation (QAE) and quantum minimum finding, to accelerate the inner minimization step of the robust Bellman update. Under this framework, the algorithm achieves a quadratic speedup in the size of the uncertainty set for the inner loop of robust RL methods. In addition to the theoretical query complexity analysis, the authors present empirical demonstrations of the proposed approach on both a quantum simulator and real quantum hardware, aiming to validate the practical feasibility of the method.

**Compliance With Llm Reviewing Policy:**

Affirmed.

**Final Justification:**

The reply is helpful in addressing most of my concerns and questions, especially those regarding the correctness of Algorithm 2. As a result, I have decided to raise my score from 2 (reject) to 3 (weak reject). The reasons I still maintain a slightly negative score are as follows:

1. I recognize that addressing the computational bottleneck of robust RL in the quantum setting is one of the main contributions of this work. However, the method appears to rely largely on a direct application of existing quantum subroutines, which limits the technical novelty of the paper. This lack of methodological originality is the main reason I am unable to give a positive score.

2. The presentation of the work needs to improve, especially the coherent process of Algorithm 2 in their paper. The current statement is quite misleading.

**Key Questions For Authors:**

**1. Clarification on the definition of the uncertainty set $\mathcal{U}$**

The paper does not provide a formal definition of the global uncertainty set $\mathcal{U}$. It is unclear whether:

- $\mathcal{U} = \prod_{(s,a)} \mathcal{U}_{sa}$ (i.e., the full rectangular product set), or
- $\mathcal{U}$ refers to a local uncertainty set $\mathcal{U}_{sa}$ at a fixed state–action pair.

Since the query complexity is stated in terms of $|\mathcal{U}|$, a precise definition is necessary. If the minimum search is performed locally at each $(s,a)$, then the relevant cardinality appears to be $|\mathcal{U}_{sa}|$, not the global product set. Please clarify the exact object whose size determines the complexity.

---

**2. Justification of discretizing $\mathcal{U}_{sa}$**

The algorithm assumes that each of it can be discretized into $N$ candidates. However, no structural assumptions (e.g., Hölder continuity, Lipschitz continuity, bounded geometry, or finite covering number) are imposed on $\mathcal{U}_{sa}$.

- Under what conditions is such a discretization legitimate?
- How does the discretization error affect the robust Bellman operator?
- Does the analysis require regularity conditions on the uncertainty set to guarantee that the discretized problem approximates the original robust problem?

Without further assumptions, it is unclear whether the discretization step is theoretically justified.

---

**3. Coherence of the `LESS` oracle in Algorithm 2 (QRIM)**

Algorithm 2 defines the function `LESS(i,j)` via classical-style calls to `ESTIMATE`, which itself invokes quantum amplitude estimation (QAE).

However, QAE involves a measurement step to obtain $\hat{g}_i$. If `ESTIMATE` returns classical information, then `LESS(i,j)` appears to:

1. Obtain classical estimates $\hat{g}_i$ and $\hat{g}_j$ via measurements,
2. Perform a classical comparison,
3. Return a phase for that specific pair $(i,j)$.

This procedure is not unitary.

Dürr–Høyer minimum finding requires a coherent unitary oracle of the form:

$$
U: \ket{i}\ket{j}\ket{0} \mapsto \ket{i}\ket{j}\ket{\mathbf{1}\\{f(i)<f(j)\\}}, \forall i,j,
$$

or

$$
U: \ket{i}\ket{j} \mapsto (-1)^{\mathbf{1}\{f(i)<f(j)\}} \ket{i}\ket{j},
\quad \forall i,j.
$$

Therefore:

- How is the `LESS` oracle implemented coherently?
- Is `ESTIMATE` performed fully coherently without intermediate measurement?

Since the query complexity analysis (e.g., Proposition 4.6) relies on Dürr–Høyer minimum finding, the validity of the complexity bound depends on `LESS` being a valid unitary oracle. Clarification on this implementation detail is essential.

---

**4. Lack of lower bounds**

The paper claims a quadratic speedup in query complexity. However, no matching lower bound is provided under the same oracle model.

- Is there a lower bound establishing that $O(\sqrt{|\mathcal{U}_{sa}|})$ is optimal?

Providing a lower bound (even under restricted assumptions) would significantly strengthen the theoretical contribution.

---

**5. Theoretical robustness advantage**

The paper empirically claims improved robustness. However, it is not theoretically explained why the quantum algorithm should yield a more robust policy than its classical counterpart beyond query efficiency.

- Does the quantum method improve robustness in a formal sense?
- Or does it only reduce the computational cost of solving the same robust Bellman equation?

Clarifying whether the contribution is algorithmic efficiency or robustness theory would help position the results more precisely.

**Limitations:**

Yes

**Strengths And Weaknesses:**

### Strengths

- To the best of my knowledge, this is the first work that explicitly studies quantum speedups in the setting of robust reinforcement learning. The focus on accelerating the inner minimization of the robust Bellman operator is well motivated and targets a genuine computational bottleneck in robust RL.

- The authors attempt to validate the theoretical claims through empirical demonstrations on both quantum simulators and real quantum hardware, which strengthens the practical relevance of the work.

---

### Weaknesses

- **Coherence of Algorithm 2 (QRIM).**
  Algorithm 2 appears to invoke quantum amplitude estimation (QAE) via classical-style `ESTIMATE` calls, which return classical approximations of $\hat{g}_i$. The subsequent `LESS` function then performs a classical comparison between $\hat{g}_i$ and $\hat{g}_j$ and applies a phase based on that comparison. As written, this procedure does not seem to define a fully coherent unitary oracle required by Dürr–Høyer minimum finding. Since the quadratic speedup relies on a valid coherent comparison oracle of the form
  $$
  U: \ket{i}\ket{j} \mapsto (-1)^{\mathbf{1}\\{f(i)<f(j)\\}} \ket{i}\ket{j}, \forall i,j,
  $$
  clarification is needed on how QAE and the comparison are implemented coherently without intermediate measurement. Without such a construction, the claimed query complexity improvement may not be theoretically justified.
- **Limited algorithmic novelty.**  The proposed approach relies entirely on existing quantum subroutines, namely quantum amplitude estimation and quantum minimum finding, without introducing new quantum algorithmic techniques. The combination of these primitives to obtain a quadratic speedup for optimization-type inner loops has been shown in prior literature (e.g., Wang et al., 2021). As a result, the technical contribution appears to be primarily an application of known quantum tools to a robust RL setting, rather than the development of fundamentally new quantum algorithmic ideas.
-  **Lack of quantum lower bounds.**
   While the paper claims a quadratic speedup in query complexity, no corresponding lower bound is provided under the same oracle model. Without a matching lower bound, it remains unclear whether the achieved complexity is optimal or whether further improvements are possible. Establishing even a restricted lower bound would significantly strengthen the theoretical contribution.

---

> ### Author Rebuttal · Authors · 2026-03-30
>
> We sincerely thank the reviewer for the and detailed feedback. We address each concern below.
>
> **Weakness 1 and Question 3**
>
> ---
> We agree that the pseudocode in Algorithm 2 (e.g., variable assignment and `return`) can be misread as requiring intermediate measurement. Our intended implementation is fully coherent, and no intermediate measurement is performed within the quantum routine.
>
> Specifically, the `ESTIMATE` operation corresponds to a coherent unitary $U_{\text{EST}}$, which encodes the estimated value $\hat{g}_i$ into an ancilla register using a fully coherent variant of quantum amplitude estimation (QAE) without measurement. The `LESS` oracle is then implemented as:
>
> 1. Apply $U_{\text{EST}}$ to obtain $|i,j\rangle|\hat{g}_i\rangle|\hat{g}_j\rangle$
> 2. Apply reversible comparator $U_{\text{CMP}}$ with controlled phase flip $(-1)^{[\hat{g}_i < \hat{g}_j]}$
> 3. Apply $U_{\text{EST}}^{\dagger}$ to uncompute auxiliary registers, restoring them to $|0\rangle$
>
> This results in the unitary $|i,j\rangle \mapsto (-1)^{[\hat{g}_i < \hat{g}_j]}|i,j\rangle$, which satisfies the requirement of Dürr–Høyer minimum finding. The only measurement in QRIM occurs at the final step, where the classical value $\hat{Q}^{\text{rob}}$ is extracted for the outer RL loop.
>
> To eliminate ambiguity, we will revise Algorithm 2 to explicitly reflect this coherent construction, and clarify the phase-kickback implementation and the absence of intermediate measurements in Section 4.2.
>
> **Weakness 2**
>
> ---
> We address this concern in detail in our response to Reviewer Hkq3 (Weakness 1).
>
> In brief, our contribution is not a new quantum primitive but a novel problem formulation: identifying the inner minimization of robust RL as a quantum search problem, with a dedicated error propagation analysis (Propositions 4.3, Theorem B.2) with additional outer-loop policy bound (Theorem B.5).
>
> **Weakness 3 and Question 4**
>
> ---
> We agree that discussing lower bounds clarifies the optimality of the proposed query complexity.
>
> In our setting, the inner minimization reduces to identifying the minimum over $N$ candidate environments given oracle access to their evaluations. Since our formulation does not assume any structural properties (e.g., convexity, smoothness, or gradient access) over the uncertainty set, this is an instance of unstructured search, which admits a quantum query lower bound of $\Omega(\sqrt{N})$ (Bennett et al., 1997).
>
> Since quantum minimum finding (Dürr–Høyer) achieves $\mathcal{O}(\sqrt{N})$ query complexity, the quadratic speedup achieved by QRIM is optimal under this model. We will add this lower bound argument to Section 4.
>
> **Question 1**
>
> ---
> The query complexity of QRIM refers to the local uncertainty set $U_{sa}$ at each state-action pair, not the global product set. QRIM performs inner minimization independently at each $(s,a)$, so the relevant cardinality is $|U_{sa}| = N$ and the per-call complexity is $\mathcal{O}(\sqrt{N})$. We will revise all complexity statements to use $|U_{sa}|$ consistently.
>
> **Question 2**
>
> ---
> We first note that our primary setting assumes $\mathcal{U}_{sa}$ is given as a finite candidate set, which is a common formulation in Robust RL. In this case, the problem is treated directly as a finite-set minimization, rather than as a discretized approximation of a continuous uncertainty set, and therefore does not require additional regularity assumptions.
> We address the reviewer's sub-questions for the case where a continuous set assumed to be discretized.
> 1. When the uncertainty set is continuous, standard regularity conditions should be considered. Lipschitz continuity of $g(s,a,u)$ with respect to the uncertainty parameter $u$ ensures that a finite grid provides a controlled approximation of the original problem.
> 2. If discretization induces a uniform error $\varepsilon_\text{disc}$, i.e., $\sup_{s,a} |\min_u g(s,a,u) - \min_i g(s,a,u_i)| \leq \varepsilon_\text{disc}$, then by standard contraction arguments, $\|V^* - \hat{V}^*\|_\infty \leq \varepsilon_\text{disc} / (1-\gamma)$. Combined with QRIM's estimation error, the total bound becomes $(\varepsilon_\text{disc} + \varepsilon_\text{cmp} + \varepsilon_\text{est})/(1-\gamma)$.
> 3. Our analysis requires no additional assumptions when $\mathcal{U}_{sa}$ is finite. Regularity conditions are only needed when interpreting the finite set as an approximation of an underlying continuous set.
>
> We will add the above discussion in Section 3
>
> **Question 5**
>
> ---
> QRIM does not introduce a new robustness objective. Our contribution is algorithmic efficiency: reducing the query complexity of the inner minimization, rather than a formal improvement in robustness.
>
> The empirical differences in Table 3 should be interpreted with care. Classical RRL computes the exact worst case, while QRIM approximates it probabilistically. This stochasticity may affect training dynamics, but we do not claim a formal robustness advantage from this effect.

---

> > ### Author Rebuttal · Reviewer_fvMD · 2026-04-01
> >
> > Thank you to the authors for the detailed response. The reply is helpful in addressing most of my concerns and questions, especially those regarding the correctness of Algorithm 2. As a result, I have decided to raise my score from 2 (reject) to 3 (weak reject). The reasons I still maintain a slightly negative score are as follows:
> >
> > 1. I recognize that addressing the computational bottleneck of robust RL in the quantum setting is one of the main contributions of this work. However, the method appears to rely largely on a direct application of existing quantum subroutines, which limits the technical novelty of the paper. This lack of methodological originality is the main reason I am unable to give a positive score.
> > 2. Although a quantum lower bound for the minimization problem can be readily obtained by extending the lower-bound argument based on Grover’s search, such a bound does not appear to be tight for the quantum robust inner minimization problem across all parameters, such as $\epsilon$.

---

> > > ### Author Response · Authors · 2026-04-05
> > >
> > > We sincerely thank the reviewer for the thoughtful reconsideration and for raising the score. We hope to share one additional perspective that motivates our work.
> > >
> > > ---
> > >
> > > ## **1. Regarding methodological originality**
> > >
> > > We acknowledge that our work uses established quantum subroutines. However, we hope to emphasize that our main goal is to discover quantum’s unseen value in advancing modern deep learning (RL, in our case). **We believe that this kind of “quantum to DL bridge” is indeed acknowledged in both societies.** In a quantum society, researchers continually seek new convergence values for quantum frameworks (e.g., their subroutines) due to the limited practicality of quantum technology. In a DL society, researchers understand quantum as a futuristic concept, but they are doubtful about why quantum is needed for the modern DL setting. **We suggest that QRIM exactly spots this point.** Specifically, QRIM raises the quantum’s new convergence value in DL (`Quantum’s desire: new value in other society, i.e., DL`) by demonstrating that quantum subroutines remarkably reduce the robust DL’s excessive query complexity (`DL’s question: Why we need quantum`)
> > >
> > > This direction of research has been appreciated in recent major ML conferences:
> > >
> > > - Wang et al. (ICML 2021): accelerate value iteration via QAE and quantum minimum finding
> > > - Ganguly et al. (NeurIPS 2023): achieve logarithmic regret in episodic MDPs via quantum exploration
> > > - Zhong et al. (NeurIPS 2024): logarithmic regret for discounted MDPs via quantum mean estimation
> > > - Xu & Aggarwal (ICML 2025): accelerate policy gradient via quantum mean estimation
> > >
> > > **All of the works use existing quantum subroutines without introducing new ones, and have been recognized as meaningful contributions to the field.** Inventing new subroutines would be huge advancement, but we believe that “discovering new value by building on the existing one” is particularly necessary in Q-DL convergence. Based on QRIM’s results, proposing new quantum subroutines tailored to QRIM scenario can be a promising future direction.
> > >
> > > ---
> > >
> > > ## **2. Regarding $\varepsilon$-tightness of the lower bound**
> > >
> > > We agree that the lower bound via BBBV addresses only the $N$-dependence. However, our formulation decomposes the inner minimization into two individually optimal primitives: (i) unstructured minimum search over $N$ candidates, with $\Omega(\sqrt{N})$ lower bound (Bennett et al., 1997), and (ii) amplitude estimation to precision $\varepsilon$, **with $\Omega(1/\varepsilon)$ lower bound from the Heisenberg limit (Giovannetti et al., 2006).** QRIM composes these sequentially, achieving $O(\sqrt{N}/\varepsilon)$. Whether exploiting the joint structure of estimation and selection could yield a tighter $\varepsilon$-dependence remains an interesting open question for future research. **Importantly, our main claim is the quadratic separation from $O(N/\varepsilon^2)$ to $O(\sqrt{N}/\varepsilon)$, which holds independently of whether the joint lower bound is tight.**
> > >
> > > **Convergence under varying $\varepsilon$:** When **$\varepsilon$** varies, it is trivial to show performance degradation due to the noisy estimation. To directly observe QRIM’s sensitivity against this, we additionally conducted ablation experiments on CartPole to examine practical convergence under increasing estimation noise, following the error model in Proposition 4.3 and Algorithm 2. Specifically, we set $\varepsilon_{\mathrm{cmp}} = \varepsilon_{\mathrm{est}} = \varepsilon$ and varied $\varepsilon \in \{0.01, 0.05, 0.10, 0.15, 0.20\}$, where $\varepsilon = 0.01$ is the value used in our main experiments (Sec. 5.4).
> > >
> > > | Metric | ε=0.01 | ε=0.05 | ε=0.10 | ε=0.15 | ε=0.20 |
> > > | --- | --- | --- | --- | --- | --- |
> > > | Train | 76.41 | 77.27 | 72.96 | 62.14 | 56.14 |
> > > | Extrap. | 58.06 | 54.34 | 52.05 | 38.23 | 27.04 |
> > >
> > > **PPO convergence remains stable, and evaluation robustness is maintained up to** $\boldsymbol{\varepsilon > 0.15}$, **indicating the practical tolerance boundary**. It is not surprising that beyond $\boldsymbol{\varepsilon > 0.15}$ lies the too-noisy regime, thereby degrading any quantum frameworks.

---

### Official Review · Reviewer_rcoY · 2026-03-12

**Soundness:** 3
**Presentation:** 2
**Significance:** 3
**Originality:** 3
**Overall Recommendation:** 3
**Confidence:** 4

**Summary:**

This paper proposes Quantum Robust Inner Minimization (QRIM), a quantum-inspired approach to accelerating the inner minimization step of robust reinforcement learning (RRL). The authors propose to encode the uncertainty set in quantum superposition and use quantum primitives, specifically amplitude estimation and quantum minimum finding, to identify the worst-case scenario more efficiently. Under this formulation, the inner minimization step can be solved with $\mathcal{O}(\sqrt{|U|})$ query complexity (rather than $\mathcal{O}(|U|)$).  The approach is evaluated in two environments (FrozenLake and CartPole) under training, interpolation, and extrapolation settings to test robustness under distribution shift. The authors provide theoretical guarantees for the method and empirically evaluate robustness, query complexity, and learning dynamics. Additional experiments include noise-model simulations and small-scale demonstrations on real quantum hardware.

**Compliance With Llm Reviewing Policy:**

Affirmed.

**Final Justification:**

This paper presents a technically sound and well-motivated approach to accelerating robust RL via quantum search. Strengths include the combination of theory and experiments, analysis of query complexity and error propagation, and the inclusion of noise-model and hardware results, which are transparently reported.

The rebuttal addressed several concerns, particularly clarifying the experimental setup, improving notation, and explicitly separating query complexity from wall-clock cost. These responses increased my confidence in the soundness and practical positioning of the work.

Regarding originality, my view is moderate. While the method builds on established quantum primitives, I agree that the paper goes beyond a trivial combination by identifying and analyzing a relevant RL bottleneck. I therefore consider the contribution borderline but sufficient for consideration, rather than clearly lacking novelty.

My remaining concern is the feasibility of incorporating the proposed revisions. Given the current density of the manuscript, I am not fully convinced that all improvements can be integrated cleanly within the page limits while resolving the presentation issues.

Overall, this is a borderline paper with clear merits, but due to uncertainty about the final presentation quality, I maintain my overall recommendation.

**Key Questions For Authors:**

1. **Computational complexity:** While QRIM improves query complexity, how does the computational cost of each oracle call compare to classical approaches when implemented via classical simulation?
2. **Experimental setup clarification:** The paper states that PPO is used in training, but the FrozenLake setup references Q-learning variants. Could the authors clarify the exact algorithms used in each experiment?
3. **Theoretical intuition:** Could the authors provide intuitive explanations for key theoretical components (e.g., Propositions 3.4, 3.5, and 4.2) in the main text?
4. **Notation and definitions:** Several variables and symbols appear without clear definitions (e.g., $T_{rob}$, $TD_{rob}$, QAE, $\mathcal{U}_{eval}$). Could the authors clarify these and ensure that all notation is introduced when first used?

**Limitations:**

Yes.

**Strengths And Weaknesses:**

## Strengths

- **Principled integration of theory and experiments.** The paper presents a coherent combination of theoretical analysis and empirical evaluation. The formulation of robust RL inner minimization as a quantum search problem is well motivated, and the resulting algorithm is accompanied by theoretical guarantees on query complexity and estimation accuracy.
- **Relevant and timely problem.** Reducing the cost of robust policy training is an important problem in reinforcement learning, especially when uncertainty sets grow large. Framing this bottleneck through the lens of quantum algorithms provides an interesting interdisciplinary perspective.
- **Empirical validation of query-efficiency benefits.** The experimental section analyzes both robustness and query-efficiency scaling, including cumulative oracle queries and learning curves measured against query counts. This helps illustrate how the theoretical speed-up may translate into faster convergence with respect to environment interactions.
- **Inclusion of hardware and noise-model experiments.** The paper includes experiments using both noise-model simulations and small-scale real quantum hardware. While the hardware results understandably do not reach the performance of noise-free simulations, the inclusion of these experiments demonstrates awareness of practical constraints and provides useful transparency regarding the feasibility of the approach.


## Weaknesses
- **Presentation.** The main limitation of the paper lies in clarity and information density, which varies considerably across sections. Some sections are overly verbose, particularly the abstract and introduction, which include informal phrasing (e.g., “our answer is Yes”, “simply mean”). Other sections lack necessary explanations or context, which makes several theoretical components difficult to follow. This uneven distribution of detail creates a harsh contrast in information density and reduces overall readability. Specific issues include:
    - Algorithm 1 provides limited additional information beyond what is already described in the text.
    - Propositions 3.4 and 3.5, while central to the method, are presented without sufficient intuition. Providing explanations in the main text and moving the formal statements and proofs to the appendix could improve readability.
    - Proposition 4.2 similarly lacks explanatory context; space used for Algorithm 1 could instead be used to provide intuition for this result.
    - Several variables and symbols are introduced without clear definitions or references, including $T_{rob}$, $TD_{rob}$, (both regarding T, TD, and the subscript rob), QAE, and $\mathcal{U}_{eval}$.
    - The statement *“For episode-wise robustness at evaluation, we fix the selected index* \hat{i} *for an entire trajectory …”* is unclear because the definition of \hat{i} is not explained.
- **Structure.** The current ordering of sections may hinder accessibility. Swapping the Related Work and Preliminaries sections could improve readability by introducing required concepts before comparing related approaches.
- **Accessibility.** The paper relies heavily on quantum computing concepts (e.g., unitaries, amplitude estimation, entanglement) but does not provide a brief introduction to these ideas. Including short explanations and intuitions, especially regarding why these principles lead to improved sample/query complexity, would significantly improve accessibility for readers not familiar with quantum computing.
- **Experimental clarity.** Tables 1 and 2 list ranges and grid counts; providing the exact sampled values rather than only ranges and counts might improve transparency. Also, there appears to be a methodological inconsistency in the experimental description: the paper states that PPO is used (l. 305, left), while the FrozenLake setup references Q-learning (l.321, right). This should be clarified.
- **Complexity analysis.** While the paper analyzes query complexity, it does not discuss computational complexity per oracle call. Given that the current implementation relies on classical simulation of quantum oracles and that NISQ devices are limited, it would be valuable to discuss the classical computational overhead of the approach, even if the number of oracle calls is reduced.

---

> ### Author Rebuttal · Authors · 2026-03-30
>
> We sincerely thank the reviewer for the thorough and constructive feedback on presentation, accessibility, and experimental clarity. We address each concern below.
>
> **Weakness 1 and Questions 3, 4**
>
> ---
> We agree that the current manuscript has uneven information density and several points where intuition or definitions could be strengthened. To address these concerns, we will make the following specific revisions:
>
> 1. **Algorithm 1:** We respectfully note that Algorithm 1 serves to show that the outer RL loop remains entirely unchanged, and QRIM acts as a drop-in replacement at line 5 of Algorithm 1 only. We believe this is worth retaining for clarity.
>
> 2. **Propositions 3.4, 3.5, 4.2:** We will add short intuitive explanations before each proposition in the main text.
>    - Proposition 3.4: QAE leverages quantum interference to estimate an expected value encoded in a quantum amplitude. While classical Monte Carlo requires $O(1/\varepsilon^2)$ samples to achieve additive error $\varepsilon$, QAE achieves the same accuracy with only $O(1/\varepsilon)$ coherent queries to the state-preparation unitary.
>    - Proposition 3.5: Quantum minimum finding exploits Grover-style amplitude amplification to search over $N$ unstructured candidates in $O(\sqrt{N})$ queries, compared to the classical $O(N)$. Combined with QAE, this enables QRIM to both estimate and compare candidate returns in superposition.
>    - Proposition 4.2: The key challenge is that the expected one-step return $g_i = \mathbb{E}[r + \gamma V(s')]$ does not appear as a single number in any quantum register, but it is distributed across branches of the superposition. The following construction resolves this by encoding $g_i$ as the probability of measuring $|1\rangle$ on an ancilla qubit, which QAE can then efficiently estimate.
>
> 3. **Notation:** We will define all symbols inline at first use by inserting short clarifications into existing sentences. We respectfully note that QAE is defined at Line 111 of the original submission.
>    - $T_{\text{rob}}$: "the robust optimality operator ($\mathcal{T}_{\text{rob}}$) becomes a Max-Min problem" (Line 138)
>    - $Q^{\text{rob}}$: "the robust action-value function $Q^{\text{rob}}(s,a;V)$ satisfies" (Proposition 3.2)
>    - $TD^{\text{rob}}$: "Value-based learners replace classical targets by the robust temporal-difference target ($TD^{\text{rob}}$)" (Line 225)
>    - $U_{\text{eval}}$: "averaged over the evaluation uncertainty set ($\mathcal{U}_{\text{eval}}$)" (Line 302)
>    - $\hat{\imath}$: "the selected index of the worst-case environment returned by the quantum minimum finding procedure" (Line 245)
>
>
>
> **Weakness 3**
>
> ---
> We agree that a brief introduction to the relevant quantum computing concepts would improve accessibility. We will add a short "Quantum Computing Basics" paragraph at the beginning of Section 3.3, which covers superposition, unitary operations, and quantum interference along with the mechanisms behind the quadratic query reductions in Propositions 3.4 and 3.5
> This paragraph, combined with the intuitive explanations added before Propositions 3.4, 3.5, and 4.2, (addressed in Weakness 1) will make the paper self-contained for readers without a quantum computing background.
>
>
>
> **Weakness 4 and Question 2**
>
> ---
> Regarding Tables 1 and 2, we will replace the current format with exact grid values used in training and evaluation (e.g., slip probability $p \in \{0.000, 0.033, 0.067, 0.100\}$), improving transparency and reproducibility.
>
> The PPO reference (Line 305) applies exclusively to CartPole, while FrozenLake uses tabular Q-learning. We will revise the sentence to clearly distinguish the two settings as follows: "...a tabular domain (FrozenLake-$8 \times 8$), where the inner loop is implemented with simulated coherent quantum routines and the outer learner is a tabular Q-learner; and a continuous-control domain (CartPole Swingup), where the inner loop uses classical emulation of quantum circuits and the outer learner is Proximal Policy Optimization (PPO)."
>
>
>
> **Weakness 5 and Question 1**
>
> ---
> We agree that, under classical simulation, each quantum oracle call incurs significant overhead due to state-vector simulation, and the wall-clock runtime of emulated QRIM exceeds that of classical baselines.
>
> However, our experimental comparison follows the standard protocol in quantum algorithm and RL research, where methods are compared based on the number of environment queries (oracle calls) rather than wall-clock time—each evaluation is treated as a unit-cost operation. Under this abstraction, QRIM reduces the required evaluations from $\mathcal{O}(N)$ to $\mathcal{O}(\sqrt{N})$, which is a genuine algorithmic improvement independent of the underlying implementation.
>
> We will clarify this in the revised manuscript by explicitly separating query complexity from per-call computational cost, and noting that our experiments validate algorithmic scaling rather than wall-clock speedup.

---

> > ### Author Rebuttal · Reviewer_rcoY · 2026-04-03
> >
> > My main technical questions have been addressed by the rebuttal, and I appreciate the authors’ constructive and detailed responses. In particular, the clarification regarding the use of Q-learning in FrozenLake and PPO in CartPole resolves an important ambiguity, and the planned additions on notation, intuition, and accessibility would substantially improve the paper. I also appreciate the explicit acknowledgment that the current implementation does not provide a wall-clock advantage under classical simulation.
> >
> >
> >
> > My remaining concern is mainly about presentation feasibility rather than technical soundness: the rebuttal proposes several useful additions, but it is unclear to me how all of them can be incorporated within the page limit without removing or condensing other content. In addition, I still believe that the paper would benefit from a more explicit discussion of classical computational overhead alongside query complexity, especially for simulated or hybrid near-term settings, to give readers a more transparent assessment of practical benefits and limitations.

---

> > > ### Author Response · Authors · 2026-04-04
> > >
> > > We thank the reviewer for the thoughtful acknowledgement. We address the remaining concerns below.
> > >
> > > ## **1. Revision content incorporation.**
> > >
> > > ---
> > >
> > > We have carefully planned the revisions to fit within the page limit, moving detailed technical material to the appendix and integrating concise explanations with conceptual diagrams. Main revisions are summarized as follows.
> > >
> > > - **Intuitive diagram and explanation of Quantum algorithms (addressing Hkq3, rcoY, fvMD):** We will add a brief "Quantum Computing Basics" paragraph and intuitive explanations before Propositions 3.4, 3.5, and 4.2 in `Section 3.3`, along with a hierarchical circuit overview figure illustrating the QRIM pipeline. We will also revise *Algorithm 2* to clarify the coherent construction of the LESS oracle in `Section 4.2`. Gate-level circuit diagrams will be placed in the Appendix.
> > > - **Positioning of the problem setting and valid baselines (addressing Hkq3, Dw5e):**  We will add a positioning paragraph in `Section 5.1` explaining why advanced classical and quantum minimax methods are inapplicable to our discrete setting, and a discussion of the discretization of $U_{sa}$ in `Section 3.2`. Additional baseline comparison results will be placed in the Appendix.
> > > - **Theoretical bounds (addressing Hkq3, fvMD):** We will add a lower bound argument ($\Omega(\sqrt{N})$, Bennett et al., 1997) in `Section 4.4`. Theorem B.5 (outer-loop policy suboptimality) and convergence-under-varying-$\varepsilon$ ablation results will be placed in the Appendix.
> > > - **Practical overhead of query complexity (addressing Dw5e, rcoY):** We will add a clarification separating query complexity from per-call computational cost in `Section 4.4`, including the precision–query tradeoff discussion. A detailed analysis of classical computational overhead across three hardware regimes will be added to the Appendix as addressed in Concern 2 below.
> > > - **Notation and presentation clarity (addressing rcoY, fvMD):** We will add inline symbol definitions ($T_{\rm rob}$, $Q^{\rm rob}$, $TD_{\rm rob}$, $U_{\rm eval}$, $\hat{i}$), exact grid values in Tables 1–2, revise sentences to clearly distinguish between the FrozenLake and CartPole experimental setups, and use $|U_{sa}|$ consistent for uncertainty set notation.
> > >
> > > These revisions directly address the reviewer’s concerns about presentation feasibility, while keeping the paper self-contained and within the page limit. We do not anticipate needing to remove or condense existing main-body content.
> > >
> > > ## **2. Classical computational overhead alongside query complexity.**
> > >
> > > ---
> > >
> > > We agree that discussing per-call computational cost alongside query complexity would give readers a more transparent assessment of practical benefits and limitations. We will add a discussion in the Appendix `analyzing the overhead across three hardware regimes:`
> > >
> > > - **Classical emulation:** Each quantum oracle call requires state-vector simulation with $O(2^n)$ operations per gate, where $n$ is the number of qubits. While the query count reduces to $O(\sqrt{N})$, the per-call cost is exponentially larger than a classical environment evaluation, and **QRIM does not yield a computational advantage in this regime**.
> > > - **NISQ (hybrid near-term):** Unlike classical emulation, the inner minimization can be executed on actual quantum hardware. However, CPU-QPU communication latency is incurred at every oracle call, adding a constant per-call overhead. For continuous environments requiring many qubits and circuit depth proportional to $O(b^2)$ per arithmetic operation, reliable execution on current noisy devices remains limited. Nevertheless, for discrete/tabular environments where the oracle circuit is shallow, NISQ execution is already feasible—as demonstrated in our FrozenLake experiment on IBM 127-qubit hardware. In such settings, **as the uncertainty set size $N$ grows, the $O(\sqrt{N})$ query reduction becomes increasingly significant.** However, fairly measuring this advantage in wall-clock time remains difficult, as the communication overhead varies across hardware platforms and integration architectures.
> > > - **Fault-tolerant:** With error-corrected quantum hardware, the per-call cost reduces to the oracle circuit's gate depth, eliminating both the exponential simulation overhead of classical emulation and the noise limitations of NISQ devices. The $O(\sqrt{N})$ query reduction then translates directly into a computational advantage, as each oracle call executes reliably at a cost determined solely by the circuit depth. **In this regime, QRIM provides an unconditional quadratic speedup over any classical method for the inner minimization.**

---

### Official Review · Reviewer_Dw5e · 2026-03-12

**Soundness:** 3
**Presentation:** 3
**Significance:** 3
**Originality:** 3
**Overall Recommendation:** 4
**Confidence:** 3

**Summary:**

This paper proposes a Quantum Robust Inner Minimization (QRIM) method to accelerate the Max-Min optimization process in Robust Reinforcement Learning (RRL). The main idea is to encode the uncertain set of the environment with quantum superposition and find out the worst case with the quantum minimum search algorithm. By this method, the query complexity is improved to $O(\sqrt{|U|})$, which is a quadratic speed-up over the classical $O(|U|)$ method, where $U$ is the environment uncertainty set. This paper validates the proposed method in a tabular environment (FrozenLake) and a continuous environment (CartPole Swingup) and some experiments are even conducted on real IBM 127-qubit hardware.

**Compliance With Llm Reviewing Policy:**

Affirmed.

**Final Justification:**

My questions are basically addressed and I would like to keep the initial rating.

**Key Questions For Authors:**

Please compare the proposed quantum method with some advanced classical method, theoretically or empirically.

Please discuss the trade-off between the precision of amplitude estimation and the query complexity.

**Limitations:**

The comparison with classical baseline is incomplete and it should consider more advanced classical methods. In addition, it lacks a deep discussion of the trade-off between the precision and the query complexity. Last but least, although the proposed method shows the potential of quantum advantage for solving robust RL problem, the advancement that reliable large scale quantum computers have not been available limits its real application.

**Strengths And Weaknesses:**

Strengths:

This paper explicitly reveals the key bottleneck of robust reinforcement learning - inner minimization requires to search through the whole uncertain set and and the cost increases dramatically when $U$ becomes large.

It makes sense to have the quantum-classical hybrid architecture design. The proposed QRIM is a plug-in module to replace the inner minimization sub-routine, and the outer RL procedure remains. This modular design has very good practicability and compatibility.

This paper also provides solid theoretical analysis, including the correctness of amplitude encoding, accuracy guarantee of QRIM, and query complexity analysis.

The experiments are also comprehensive. It validates the quadratic speed-up in the classical simulator and verifies the robust in real IBM 127-qubit quantum device.

Weaknesses:

The comparison with classical baseline is incomplete. This paper only compares with the  classical baseline of naive search method and ignores some advanced classical methods such as some adaptive sampling strategies or robust RL based on convex optimization.

The trade-off between the precision of amplitude estimation and query complexity should be further discussed. It seems that it requires a large amount of queries when the precision parameter $\epsilon$ is very small. In this case, is the quantum method still better than the classical methods?

---

> ### Author Rebuttal · Authors · 2026-03-30
>
> We sincerely thank the reviewer for the constructive and detailed feedback. We address each concern below.
>
> **Weakness 1 and Question 1**
>
> ---
>
> We agree that a fairer comparison should involve advanced randomized classical methods rather than purely deterministic exhaustive search. Our inner minimization operates over a discrete, unstructured finite set of $N$ candidates, where no convexity, smoothness, or ordering is assumed among $\{g_i\}$. This setting reflects the practical reality that we do not assume differentiability or convexity with respect to the uncertainty parameters, and access each candidate environment only through evaluations. In this setting, continuous stochastic minimax solvers (e.g., stochastic mirror descent) are not directly applicable due to the lack of gradient access and continuous parameterization.
>
> The most relevant classical randomized alternative is best-arm identification, which adaptively samples to identify the worst-case candidate with high-probability guarantees. We have conducted additional experiments comparing QRIM against randomized classical methods under identical settings of FrozenLake as Table 3 and 4.
>
> | Method       | Queries | Train       | Interp.     | Extrap.      |
> |--------------|---------|-------------|-------------|--------------|
> | Classic RRL  | 11.45M  | 0.84 ± 0.06| 0.81 ± 0.06 | −0.45 ± 0.17 |
> | BAI-RRL      | 9.67M   | 0.96 ± 0.02| 0.95 ± 0.02 | −0.49 ± 0.07 |
> | QRIM (ours)  | 4.11M   | 0.98 ± 0.03| 0.98 ± 0.03 | −0.37 ± 0.06 |
>
> The results confirm that QRIM maintains significant query reduction against best-arm identification (BAI-RRL) while achieving comparable robustness. BAI-RRL reduces queries by 15.5% over Classic RRL but remains O(N)-scale, while QRIM achieves 64.1% reduction with O(√N) scaling and stronger robustness across all evaluation splits.
> We will add the results evaluating both query complexity and robustness in Section 5 and Appendix.
>
> **Weakness 2 and Question 2**
>
> ---
>
> The key point lies in how the query complexity scales with both the uncertainty size $N$ and the precision parameter $\varepsilon$. The classical counterpart requires $\mathcal{O}(N / \varepsilon^2)$ queries, as it must estimate each candidate return via sampling. In contrast, QRIM combines quantum minimum finding with amplitude estimation, achieving $\tilde{\mathcal{O}}(\sqrt{N} / \varepsilon)$ query complexity.
>
> Therefore, while both methods incur higher cost as higher precision (smaller $\varepsilon$) is required, the scaling differs fundamentally. The classical method depends linearly on $N$ and quadratically on $1/\varepsilon$, whereas QRIM depends on $\sqrt{N}$ and linearly on $1/\varepsilon$. As a result, the quadratic advantage in $N$ is preserved across all precision regimes. In particular, even as $\varepsilon$ decreases, QRIM maintains a strictly better asymptotic scaling than classical methods.
>
> We will additionally clarify this tradeoff in the revised manuscript, in Section 4 and Appendix B.
>
>
>
> **Weakness 3**
>
> ---
>
> We agree that the full practical realization of QRIM awaits fault-tolerant quantum hardware. As discussed in Section 7, our current contribution focuses on establishing the algorithmic framework and complexity advantage. Meanwhile, we have demonstrated hardware feasibility for discrete environments on IBM 127-qubit devices (Section 6), providing evidence that the approach is physically realizable within current hardware limits for tractable problem sizes.

---

> > ### Author Rebuttal · Reviewer_Dw5e · 2026-04-07
> >
> > Thank the authors for the detailed response. My questions are basically addressed. The authors justified the unavailability of some advanced classical methods and provided analysis on query complexity.

---

### Official Review · Reviewer_Hkq3 · 2026-03-13

**Soundness:** 3
**Presentation:** 3
**Significance:** 3
**Originality:** 2
**Overall Recommendation:** 4
**Confidence:** 4

**Summary:**

The authors introduce Quantum Robust Inner Minimization to alleviate the computational bottleneck of the inner minimization step in robust reinforcement learning. Classical robust reinforcement learning evaluates worst case transitions sequentially, incurring a query complexity of $\mathcal{O}(|\mathcal{U}|)$. The proposed algorithm utilizes a quantum accessible environment to encode the uncertainty set into a quantum superposition. It combines standard quantum amplitude estimation and the Dürr and Høyer minimum finding algorithm to locate the most adverse backup with a reduced query complexity of $\mathcal{O}(\sqrt{|\mathcal{U}|})$. The authors prove that the approximate robust Bellman operator maintains its fixed point stability despite quantum estimation errors. Empirically, the method demonstrates query reductions while maintaining robust policy performance in both a discrete tabular environment and a continuous control environment.

**Compliance With Llm Reviewing Policy:**

Affirmed.

**Final Justification:**

Soundness (3 → 3). The theoretical formulation remains mechanically correct. The authors confirmed QRIM achieves optimal O(√N) query complexity for the discrete inner minimization, matching the Ω(√N) quantum lower bound for unstructured search (Bennett et al., 1997). The promised Theorem B.5, bounding outer-loop policy suboptimality at O(ε), strengthens the error propagation analysis by connecting quantum estimation error to final policy performance. The convergence ablation under varying ε is a direct and welcome response to our concern about PPO stability under quantum noise: training remains stable at ε = 0.10 (~5% degradation) with significant decline at ε = 0.15 (~19% training, ~34% extrapolation), establishing a practical tolerance margin. Learning curves and run-to-run variance would further solidify this finding in revision. The gate depth estimate for CartPole (~200-250 qubits, ~2,000-3,000 gates at 8-bit precision) provides the quantitative grounding we requested. However, the NISQ feasibility claim needs tempering: at a representative 0.5% two-qubit gate error rate, 2,000 gates yield (0.995)^2000 ≈ 4.4 × 10⁻⁵ circuit fidelity, which is negligible. The FrozenLake hardware experiment uses far shallower circuits and is genuinely feasible; CartPole-scale circuits are not. The wall-clock crossover condition also deserves explicit statement: classical cost is O(N), quantum cost is O(√N · b²), yielding the crossover N > b⁴. At b = 8, this requires N > 4,096 candidate environments before quantum execution outperforms classical.

Originality (2 → 2). The rebuttal reinforced that the underlying quantum primitives (QAE, Dürr-Høyer minimum finding) are established techniques. The originality lies in identifying the robust RL inner minimization as a quantum search problem and analyzing how estimation errors propagate through the robust Bellman operator. The authors' survey of classical and quantum minimax alternatives confirmed that methods requiring continuity, smoothness, or convexity are inapplicable to their discrete unstructured setting, validating the problem formulation as a genuine contribution. The theoretical machinery, however, remains a direct composition of known tools. The contribution is architectural rather than algorithmic: a novel problem mapping, not a novel algorithm.

Significance (2 → 3). The rebuttal improved significance in two respects. First, the positioning against both classical and quantum alternatives, together with the Ω(√N) lower bound, establishes that QRIM is optimally efficient for its setting. Second, the convergence ablation and gate estimates begin bridging query complexity theory and practical feasibility. One scope consideration merits acknowledgment: the discrete finite uncertainty set occupies a specific niche within robust RL. The dominant formulation uses continuous divergence balls (KL, total variation, Wasserstein), where the inner minimization is convex and admits efficient analytical solutions (Iyengar, 2005; Nilim & El Ghaoui, 2005). The bottleneck QRIM addresses does not arise there. The discrete formulation does appear in the literature: Nilim & El Ghaoui (2005) treat it as a special case, Dimitrova et al. (2016) reduce polytopic uncertainty to vertex enumeration, and Sester & Decker (2024, arXiv:2407.04259) develop Q-learning for finite ambiguity sets with linear-in-N cost. Positioning QRIM within this landscape would clarify where the quadratic speedup applies.

Presentation (3 → 3). The promised additions (circuit diagrams, Theorem B.5, intuitive explanations, inline notation) would improve accessibility substantially. One refinement is needed: the proposed Section 5.1 paragraph claims no classical method applies beyond exhaustive enumeration. This overlooks BAI, which the authors themselves adopted as a baseline and which operates on discrete, finite sets without smoothness, convexity, or gradient access. BAI does not break O(N) scaling (15.5% query reduction vs QRIM's 64.1%), so the quantum advantage holds. The precise claim should be: classical methods for this setting exist and reduce constant factors, but cannot achieve sub-linear scaling. This framing is both accurate and stronger.

Overall. The authors engaged constructively across two rounds. Core Concern 1 (positioning against alternatives) is substantively resolved. Core Concern 2 (theory-practice gap) is partially addressed with useful new evidence. We adjust from 3 (weak reject) to 4 (weak accept), contingent on incorporating the suggested revisions.

**Key Questions For Authors:**

**Key Questions For Authors***

1. **Classical Randomized Baselines:** Quantum algorithms output probabilistic results. While you compare against classical Monte Carlo bounds of $\mathcal{O}(N / \epsilon^2)$, how does the quantum query complexity compare against more advanced classical randomized optimization or stochastic gradient methods for minimax problems? A comparison against randomized classical oracles would provide a fairer baseline for demonstrating true quantum advantage.
2. **Alternative Quantum Minimax Algorithms:** You restrict the uncertainty set to a discrete grid of size $N$ to apply the minimum search. Did you consider other quantum algorithms tailored for minimax or saddle point problems? Could these alternatives bypass the discrete grid assumption or offer different complexity tradeoffs?
3. **Emulation versus Execution Overhead:** The CartPole emulation bypasses the construction of reversible arithmetic circuits by calculating amplitudes via a classical physics engine. How does the theoretical gate depth of uncomputing these continuous physics steps scale, and does this compilation overhead negate the quadratic query acceleration in practical execution time?
4. **Oracle Realizability:** The method relies strictly on the availability of a quantum accessible environment that provides coherent unitaries. Given the difficulty of modeling arbitrary transition kernels as quantum circuits, what specific classes of real world environments natively support this interface?
5. **Impact of Quantum Variance on Policy Convergence:** Classical robust reinforcement learning provides deterministic and exact targets for the worst case scenario. In contrast, quantum amplitude estimation is inherently probabilistic, introducing inherent variance and estimation errors into the Bellman targets. While Theorem B.2 bounds the theoretical fixed point error, modern actor critic methods like Proximal Policy Optimization are notoriously sensitive to noisy advantage estimates. Could the authors provide empirical or theoretical insights into how this quantum induced variance impacts the stability and sample efficiency of the outer training loop? If the outer policy gradient struggles to converge due to noisy targets, the practical training time could increase, potentially negating the quantum query acceleration entirely.

**Limitations:**

Yes. The authors transparently state that a full quantum implementation for complex environments requires deep arithmetic circuits exceeding current hardware capabilities. They also adequately acknowledge the hybrid classical quantum communication overheads. It would be beneficial for the authors to explicitly acknowledge in the main text that the theoretical results are direct applications of standard reinforcement learning and quantum bounds rather than novel theoretical contributions.

**Strengths And Weaknesses:**

**Strengths And Weaknesses***

**General Comments**
The paper presents a mechanically sound application of quantum minimum finding to the maximization minimization optimization problem in robust reinforcement learning. However, to maximize the impact of the paper, the authors should address the gap between the theoretical formulation, which relies heavily on standard mathematical bounds, and the physical realizability of the proposed quantum oracles. Furthermore, strengthening the baselines to include classical randomized algorithms and discussing alternative quantum minimax solvers would make the evaluation much more rigorous.

**Specific Comments**

* **Soundness:**
* *Strength:* The theoretical formulation is mechanically correct. The authors successfully combine known quantum bounds and standard reinforcement learning proofs. Specifically, Theorem B.2 correctly applies the standard contraction mapping property to demonstrate that fixed point perturbations caused by quantum estimation errors remain bounded.
* *Weakness 1 (Lack of Theoretical Novelty):* While the proofs are correct, the theoretical results are standard derivations in reinforcement learning and quantum computing. Theorem B.2 is a routine bounded error analysis for Bellman operators, and the query complexity bounds simply inherit the established properties of the minimum finding algorithm. The paper lacks a novel theoretical breakthrough beyond this mechanical synthesis.
* *Weakness 2 (Classical Emulation for Continuous Control):* The empirical soundness for the CartPole environment is limited by its reliance on a classical emulation strategy. Because encoding complex differential equations into reversible quantum arithmetic circuits exceeds current hardware capabilities, the CartPole experiment bypasses logic gate operations. Instead, it uses a classical physics engine to compute exact state vectors and manually injects the amplitudes mathematically. This makes the continuous control evaluation an exercise in theoretical query counting rather than a demonstration of quantum feasibility.
* *Weakness 3 (Fairness of Classical Baselines):* Quantum minimum finding is inherently probabilistic. While the authors compare their complexity against classical Monte Carlo, a more rigorous and fair comparison would evaluate the quantum approach against advanced classical randomized optimization or stochastic algorithms designed specifically for minimax problems. Comparing a probabilistic quantum oracle primarily to deterministic exhaustive scans or basic Monte Carlo may overstate the practical algorithmic advantage.


* **Presentation:**
* *Strength:* The manuscript is logically structured, clearly distinguishing between classical and quantum query counts in the experimental evaluation to illustrate the asymptotic scaling. Appendix C provides a highly necessary and transparent distinction between the coherent logic circuits used for FrozenLake and the classical emulation used for CartPole.
* *Weakness:* While the modular nature of the algorithm is emphasized, the visual presentation does not sufficiently illustrate the deep arithmetic circuits required to realize the necessary unitaries. Including a conceptual circuit diagram for the quantum accessible environment in the main text would clarify the immense compilation overhead currently abstracted away by the classical emulation.


* **Significance:**
* *Strength:* The framework addresses a severe scaling bottleneck in robust reinforcement learning, where the inner minimization cost grows rapidly with the resolution of the uncertainty set. The method acts as a modular routine that preserves the outer optimization without altering existing training pipelines, ensuring compatibility with standard algorithms like Proximal Policy Optimization.
* *Weakness:* The assumption of perfect noiseless quantum oracles for complex physics engines limits the immediate practical significance of the continuous control results. The authors admit that a full quantum implementation for complex environments would require circuits that exceed current hardware capabilities.


* **Originality:**
* *Strength:* The work successfully maps the specific robust Bellman backup equation to a quantum accessible oracle state preparation.
* *Weakness:* The underlying quantum algorithms are established techniques. The reinforcement learning theory relies on standard contraction mapping. The originality lies strictly in the architectural translation to the inner loop of the reinforcement learning process, rather than the invention of a new theoretical paradigm.

---

> ### Author Rebuttal · Authors · 2026-03-30
>
> We sincerely thank the reviewer for the thorough and constructive feedback. We address each concern below.
>
>
>
> **Weakness 1, 6, and Question 5**
>
> ---
> We agree that our work builds upon established quantum primitives and standard contraction mapping. We hope to emphasize, however, that the core novelty lies in formulating and addressing a key bottleneck in Robust RL, the exhaustive inner minimization of the Max-Min robust Bellman backup, as a quantum search problem, and rigorously bounding the resulting error propagation.
>
> Prior QRL works address standard MDPs under a single, fixed environment. Our work is first to identify the RRL inner minimization as a quantum minimum finding problem achieving $O(\sqrt{|\mathcal{U}_{sa}|})$ speedup, and analyze how the estimation errors propagate through the robust Bellman operator.
>
> We respectfully note that our contribution targets the inner minimization of robust RL, not the inner loop of standard RL.
>
> To further strengthen our theoretical contribution and directly address the reviewer's concern about PPO's sensitivity to noisy targets, we will add Theorem B.5 (Outer Loop Policy Suboptimality), covering both frameworks used in our experiments. proving that both value-based and policy-gradient learners converge under QRIM noise with suboptimality bounded by $O(\varepsilon/(1-\gamma)^2)$. which directly connects quantum estimation error to final policy performance for the first time in the QRL literature.
>
>
>
>
> **Weakness 2, 5, and Question 3**
>
> ---
> We agree that compiling continuous control dynamics (e.g., CartPole) into reversible quantum circuits would incur substantial gate depth overhead. In the current NISQ regime, this overhead would negate the quadratic query advantage in terms of wall-clock execution time. However, in the fault-tolerant regime, the per-call compilation overhead is a polynomial factor that does not affect the asymptotic query scaling.
>
> Our work follows the standard quantum-accessible oracle model, where the primary objective is to analyze and validate algorithmic query complexity rather than immediate hardware feasibility. This paradigm is widely adopted in quantum RL research where classical emulation is used to study asymptotic scaling. Our work goes a step further by additionally demonstrating hardware feasibility on real quantum devices—an aspect absent from prior quantum RL studies.
>
> Accordingly, our experiments serve two complementary goals:
>
> - CartPole (algorithmic validation): We use classical emulation to isolate the query complexity reduction and verify that the $O(\sqrt{|\mathcal{U}|})$ scaling integrates with continuous RL (PPO). The goal is to validate asymptotic behavior, not hardware feasibility.
> - FrozenLake (hardware feasibility): We construct the full coherent quantum circuit at the gate level (Appendix C.1) and execute it on real IBM hardware, demonstrating that QRIM is physically realizable within current hardware limits.
>
> Regarding gate depth, for tabular environments the circuit depth scales polynomially in the problem size (e.g., $\log|\mathcal{S}|$) and remains tractable. For continuous dynamics, a full reversible implementation remains beyond current capabilities, as acknowledged in Section 7.
>
>
>
> **Weakness 3 and Question 2**
>
> ---
> Due to the character limit, we refer to our response to Reviewer Dw5e (Weakness 1) for further details.
> In Brief, we do not assume differentiability or convexity with respect to the uncertainty parameters, and access each candidate environment only through evaluations. Therefore, optimization-based minimax methods are not directly applicable.
> we adopt Best-Arm Identification (BAI) as the classical baseline for our discrete setting, and the results confirm that QRIM maintains significant query reduction against this stronger baseline while achieving comparable robustness.
>
>
> **Weakness 4**
>
> ---
> We agree that a visual circuit diagram would improve clarity. In the revised manuscript, we will add: (1) a hierarchical overview figure in the main text illustrating both the quantum-accessible environment oracles ($U_P$, $U_R$, $U_V$) and how they compose into the full QRIM pipeline ($G_i(s,a)$ → QAE → coherent LESS comparator → Dürr–Høyer → final estimation); and (2) gate-level circuit diagrams in Appendix C for each component.
>
>
>
>
>
> **Question 4**
>
> ---
> We agree that environments natively supporting coherent quantum interfaces are limited. In practice, our framework applies to: (1) discrete/tabular environments, where transition and reward tables can be implemented with reversible circuits, as demonstrated in our FrozenLake experiment; and (2) parametric or simulator-based environments with known dynamics, which can be embedded into reversible arithmetic circuits. Truly native realizations arise in domains where the environment itself is a quantum system, which is beyond the scope of this work.

---

> > ### Author Rebuttal · Reviewer_Hkq3 · 2026-04-01
> >
> > We thank the authors for their thoughtful and detailed rebuttal. The proposed additions, Theorem B.5 and the circuit diagrams, are constructive, and the candid discussion of hardware limitations is appreciated. We select option (c) because the unresolved concerns are foundational and require revisions beyond a short rebuttal.
> >
> > As a minor note, the rebuttal groups "Weakness 1, 6, and Question 5" together, but we are unable to identify which of our concerns corresponds to "Weakness 6." We ask the authors to clarify which specific point they intended to address.
> >
> > Core Concern 1: Quantum Advantage Lacks Rigorous Positioning (Q1, Q2). The paper claims a quadratic speedup, yet compares QRIM against a narrow set of baselines. Adding BAI strengthens the classical side, but advanced classical stochastic methods for black-box minimax problems remain unexamined. More critically, the rebuttal does not address our question about alternative quantum algorithms for minimax or saddle-point problems. It instead argues against classical optimization methods, which was not our concern. A convincing speedup claim demands thorough positioning against both classical and quantum alternatives. This requires a literature survey and potentially new comparisons, work that exceeds a rebuttal's scope.
> >
> > Core Concern 2: Theory and Practice Remain Disconnected for Continuous Control (Q3, Q5). Two gaps undermine the continuous control contribution. First, the gate depth for reversible compilation of continuous dynamics remains unquantified. The rebuttal acknowledges the overhead but offers no estimate, leaving readers unable to judge when the query advantage translates into wall-clock gains. Second, the promised Theorem B.5 bounds policy suboptimality at O(epsilon), but a worst-case bound alone does not capture practical convergence. Actor-critic methods respond to noise empirically, not asymptotically. Convergence studies under varying estimation error levels would address this directly. Both points demand substantial new analysis or experiments.
> >
> > The core idea, recasting robust RL inner minimization as a quantum search problem, is genuinely interesting. Our concerns bear not on correctness but on the depth of evidence the claims require. Addressing these points in a revised manuscript would strengthen the contribution considerably.

---

> > > ### Author Response · Authors · 2026-04-04
> > >
> > > We thank the reviewer for the detailed acknowledgement and for clarifying the two core concerns. Regarding the minor note, "Weakness 6" in our response referred to the weakness in the Originality section, which appeared as the sixth item of the total weakness.
> > >
> > > ## **Core Concern 1: Positioning against classical and quantum alternatives**
> > >
> > > ---
> > > We apologize that the cross-reference to Reviewer Dw5e may have been insufficient. To address the reviewer's concern, **we reviewed advanced classical stochastic methods for black-box minimax problems and quantum algorithms for minimax or saddle-point problems**.
> > >
> > > **Classical:**
> > > - ZO-Min-Max (Liu et al., 2020)
> > > - Derivative-free minimax via ES (Al-Dujaili et al., 2018)
> > > - Gradient-free saddle-point method (Beznosikov et al., 2020)
> > >
> > > **Quantum:**
> > > - Quantum matrix game solver (Li et al., 2021)
> > > - Quantum zero-sum game learner (Gao et al., 2023)
> > > - HAQZO (Liu et al., 2025)
> > >
> > > These methods all require that the uncertainty set is continuously parameterized with smoothness, convexity of the objective, or at minimum gradient-accessible structure. `Our setting, however, gradient or convexity structure is not available, making the above methods inapplicable—reflecting the practical reality of robust RL`**:** environment dynamics change across distinct candidates rather than varying continuously over a smooth, convex parameter space, where the agent needs to perform well under the worst case among a set of candidate environments, which is widely adopted in robust RL (Nilim&El Ghaoui, 2005, Rajeswaran et al., 2017, Ho et al., 2018).
> > >
> > > **To address this in the revised manuscript, we will add the following paragraph to Section 5.1:**
> > >
> > > “Various classical/quantum algorithms have been developed for minimax problems. However, these methods require smoothness, convexity, or continuous parameterization for gradient estimation—assumptions that conflict with our setting where the inner minimization operates over a finite discrete candidate set with no such structure. Under this setting, the inner minimization reduces to unstructured search with $\Omega(N)$ classical and $\Omega(\sqrt{N})$ quantum lower bounds, in which QRIM achieves the optimal query complexity.”
> > >
> > > ## **Core Concern 2: Theory-practice gap for continuous control**
> > >
> > > ---
> > > **Gate depth estimate:** We provide an estimate for implementing a CartPole dynamics step as a reversible quantum circuit. The CartPole state consists of 4 continuous variables, represented in $b$-bit fixed-point precision. **The dynamics involve multiplications, divisions, and trigonometric evaluations, each requiring $O(b^2)$** reversible gates, plus uncomputation. At $b = 8$ bits: state register $4 \times 8 = 32$ qubits, ~200 ancilla qubits for intermediate computations, `totaling ~200–250 qubits and ~2,000–3,000 gates.` With the maximum current NISQ resource, it may be possible to implement, but we hope to excuse that beyond-hundreds qubit resource is largely intractable nowadays. Although the existing developments of quantum computers are not sufficient, we desire to raise the quadratic data-efficiency of quantum-based environment for envisioning the future value.
> > >
> > > **Wall-clock crossover:** In the fault-tolerant regime, the quantum wall-clock advantage emerges **when the query reduction $O(\\sqrt{N})$ outweighs the per-call gate overhead $O(b^2)$**, when the uncertainty set size $N$ is sufficiently larger than the per-call circuit depth, which varies with the complexity of the environment dynamics. However, our current implementation is a classical-quantum hybrid. **In this setup, wall-clock time reflects the classical simulation overhead of quantum circuits rather than the actual quantum execution cost, making a fair wall-clock comparison limited by hardware dependency rather than algorithmic factors.** Meaningful wall-clock benchmarking would require either a seamless classical-quantum platform that avoids communication overhead, or a fault-tolerant quantum processor where per-query cost is determined by gate depth. As neither currently exists, our work primarily targets query complexity—the standard evaluation metric in quantum algorithm research.
> > >
> > > **Convergence under varying $\varepsilon$:** To directly address the concern about practical convergence, **we conducted ablation experiments on CartPole**, following the error model in Proposition 4.3 and Algorithm 2. Specifically, we set $\varepsilon_{\mathrm{cmp}} = \varepsilon_{\mathrm{est}} = \varepsilon$ and varied $\varepsilon \in \{0.01, 0.05, 0.10, 0.15, 0.20\}$, where $\varepsilon = 0.01$ is the value used in our main experiments (Sec. 5.4).
> > >
> > > | Metric | ε=0.01 | ε=0.05 | ε=0.10 | ε=0.15 | ε=0.20 |
> > > | --- | --- | --- | --- | --- | --- |
> > > | Train | 76.41 | 77.27 | 72.96 | 62.14 | 56.14 |
> > > | Extrap. | 58.06 | 54.34 | 52.05 | 38.23 | 27.04 |
> > >
> > > As shown above, **PPO convergence remains stable and evaluation robustness is maintained up to $\\varepsilon = 0.10$–$0.15$, confirming a substantial tolerance margin.**

---

### Decision · Program_Chairs · 2026-04-30

**Decision:**

Accept (regular)

**Comment:**

The reviewers unanimously agree that this paper targets a highly relevant computational bottleneck in Robust Reinforcement Learning by cleverly framing the inner minimization step as a quantum search problem, achieving a theoretically sound and optimal quadratic speed-up. I have carefully read the authors' extensive rebuttals and appreciate their detailed engagement, particularly in clarifying the discrete unstructured setting, providing concrete gate depth estimates, and outlining valuable additions such as Theorem B.5 and a coherent circuit presentation. However, significant debate remains among the reviewers regarding whether the direct application of existing quantum primitives constitutes sufficient algorithmic novelty. Furthermore, while the theoretical contribution is clear, several reviewers retain serious concerns about the substantial gap between the continuous control theory and current hardware feasibility, as well as the practical challenge of integrating the sheer volume of promised, necessary revisions into the final manuscript without compromising its clarity and density.